# Numerically Accurate Hyperbolic Embeddings Using Tiling-Based Models

**Tao Yu**
Department of Computer Science
Cornell University
Ithaca, NY, USA
tyu@cs.cornell.edu

**Christopher De Sa**
Department of Computer Science
Cornell University
Ithaca, NY, USA
cdesa@cs.cornell.edu

## Abstract

Hyperbolic embeddings achieve excellent performance when embedding hierarchical data structures like synonym or type hierarchies, but they can be limited by numerical error when ordinary floating-point numbers are used to represent points in hyperbolic space. Standard models such as the Poincaré disk and the Lorentz model have unbounded numerical error as points get far from the origin. To address this, we propose a new model which uses an integer-based tiling to represent *any* point in hyperbolic space with provably bounded numerical error. This allows us to learn high-precision embeddings without using BigFloats, and enables us to store the resulting embeddings with fewer bits. We evaluate our tiling-based model empirically, and show that it can both compress hyperbolic embeddings (down to 2% of a Poincaré embedding on WordNet Nouns) and learn more accurate embeddings on real-world datasets.

## 1   Introduction

In the real world, valuable knowledge is encoded in datasets with hierarchical structure, such as the IBM Information Management System to describe the structure of documents, the large lexical database WordNet [14], various networks [8] and natural language sentences [24, 5]. It is challenging but necessary to embed these structured data for the use of modern machine learning methods. Recent work [11, 26, 27, 7] proposed using *hyperbolic spaces* to embed these structures and has achieved exciting results. A hyperbolic space is a manifold with constant negative curvature and endowed with various geometric properties, in particular, Bowditch [4] shows that any finite subset of an hyperbolic space looks like a finite tree according to the definition in [18]. Therefore, the hyperbolic space is well suited to model hierarchical structures.

A major difficulty that arises when learning with hyperbolic embeddings is the numerical instability, sometimes informally called "the NaN problem". Models of hyperbolic space commonly used to learn embeddings, such as the Poincaré ball model [26] and the Lorentz hyperboloid model [27], suffer from significant numerical error caused by floating-point computation and amplified by the ill-conditioned Riemannian metrics involved in their construction. To address this when embedding a graph, one technical solution exploited by Sarkar [32] is to carefully scale down all the edge lengths by a factor before embedding, then recover the original distances afterwards by dividing by the factor. However, this scaling increases the distortion of the embedding, and the distortion gets worse as the scale factor increases [30]. Sala et al. [30] suggested that, to produce a good embedding in hyperbolic space, one can either increase the number of bits used for the floating-point numbers or increase the dimension of the space.

While these methods can greatly improve the accuracy of an embedding empirically, they come with a computational cost, and the floating-point error is still unbounded everywhere. Despite these previous

adopted methods, as points move far away from the origin, the error caused by using floating-point numbers to represent them will be unbounded. Even if we try to compensate for this effect by using BigFloats (non-standard floating-point numbers that use a large quantity of bits), no matter how many bits we use, there will always be numerical issues for points sufficiently far away from the origin. No amount of BigFloat precision is sufficient to accurately represent points *everywhere* in hyperbolic space.

To address this problem, it is desirable to have a way of representing points in hyperbolic space that: (1) can represent any point in the space with small fixed bounded error; (2) supports standard geometric computations, such as hyperbolic distances, with small numerical error; and (3) avoids potentially expensive BigFloat arithmetic.

One solution is to avoid floating-point arithmetic and do as much computation as possible with integer arithmetic, which introduces no error. To gain intuition, imagine solving the same problem in the more familiar setting of the Euclidean plane. A simple way to construct a constant-error representation is by using the integer-lattice square tiling (or tessellation) [9] of the Euclidean plane. With this, we can represent any point in the plane by (1) storing the coordinates of the square where the point is located as integers and (2) storing the coordinates of the point within that square as floating point numbers. In this way, the worst-case representation error (Definition 1) will only be proportional to the machine epsilon of the floating-point format—but not the distance of the point from the origin.

We propose to do the same thing in the hyperbolic space: we call this a *tiling-based model*. Given some tiling of hyperbolic space, we can represent a point in hyperbolic space as a pair of (1) the tile it is on and (2) its position within the tile represented with floating point coordinates. In this paper, we show how we can do this, and we make the following contributions:

- We identify tiling-based models for both the hyperbolic plane and for higher-dimensional hyperbolic space in various dimensions. We prove that the representation error (Definition 1) is bounded by a fixed value, further, the error of computing distances and gradients are independent of how far the points are from the origin.
- We show how to compute efficiently over tiling-based models, and we offer algorithms to compress and learn embeddings for real-world datasets.

The reminder of this paper is organized as follows. In Section 2, we discuss related work regarding hyperbolic embeddings on various models. In Section 3, we detail the standard models of hyperbolic space which we use in our theory and experiments. In Section 4, we introduce the $L$-tiling model and show how it can be used to accurately represent any point in the hyperbolic plane (2-dimensional hyperbolic space). In Section 5, we show how to use the $L$-tiling model to learn embeddings with traditional manifold optimization algorithms. In Section 6, we develop the $H$-tiling model, which generalizes our methods to higher dimensional spaces. Finally, in Section 7, evaluate our methods on two different tasks: (1) compressing a learned embedding and (2) learning embeddings on multiple real-world datasets.

## 2 Related Work

Hyperbolic space [1] is a simply connected Riemannian manifold with constant negative (sectional) curvature, which is analogous to a high dimensional sphere with constant positive curvature. The negative-curvature metric of the hyperbolic space results in very different geometric properties, which makes it widely employed in many settings. One noticeable property is the volume of the ball in hyperbolic space: it increases exponentially with respect to the radius (for large radius), rather than polynomially as in the Euclidean case [6]. For comparison to hierarchical data, consider a tree with branching factor $b$, where the number of leaf nodes increases exponentially as the tree depth increases [27], this property makes hyperbolic space particularly well suited to represent hierarchies.

Nickel and Kiela [26] introduced the Poincaré embedding for learning hierarchical representations of symbolic data, which captured the attention of the machine learning community. The Poincaré ball model of hyperbolic space was used to embed taxonomies and graphs with state-of-the-art results in link prediction and lexical entailment. Similarly, it was also proposed in [7] to learn neural embeddings of graphs in hyperbolic space, where the performances on downstream tasks were improved significantly. The Poincaré ball model was used in several subsequent works, including

unsupervised learning of word and sentence embeddings [35, 13], directed acyclic graph embeddings and hierarchical relations learning using a family of nested geodesically convex cones [16]. Further, Ganea et al. [15] proposed hyperbolic neural networks to embed sequential data and perform classification based on the Poincaré ball model.

In a later work [27], the Poincaré model and the Lorentz model of hyperbolic space were compared to learn the same embeddings, and the Lorentz model was observed to be substantially more efficient than the Poincaré model to learn high-quality embeddings of large taxonomies, especially in low dimensions. Similarly, Gulcehre et al. [20] built the new hyperbolic attention networks on top of the Lorentz model rather than the Poincaré model. Further along this direction, Gu et al. [19] explored a product manifold combining multiple copies of different model spaces to get better performance on a range of datasets and reconstruction tasks. This suggests that the numerical model used for learning embeddings can have significant impact on its performance. Sala et al. [30] analyzed the tradeoffs between precision and dimensionality of hyperbolic embeddings to show this is a fundamental problem when using float arithmetic. More broadly, different models have been used in different tasks like hierarchies embedding [26], text embedding [35, 13] and question answering system [34]. However, all these models can be limited by numerical precision issues.

## 3 Models of Hyperbolic Space

Typically, people work with hyperbolic space by using a *model*, a representation of hyperbolic space within Euclidean space. There exists multiple important models for hyperbolic space, most notably the Poincaré ball model, the Lorentz hyperboloid model, and the Poincaré upper-half space model [1], which will be described in this section. These all model the same geometry in the sense that any two of them can be related by a transformation that preserves all the geometrical properties of the space, including distances and gradient [6]. Generally, one can choose whichever model is best suited for a given task [27].

**Poincaré ball model.** The Poincaré ball model is the Riemannian manifold $(\mathcal{B}^n, g_p)$, where $\mathcal{B}^n = \{\boldsymbol{x} \in \mathbb{R}^n : \|\boldsymbol{x}\| < 1\}$ is the open unit ball. The metric and distance on $\mathcal{B}^n$ are defined as

$$g_p(\boldsymbol{x}) = \left(\frac{2}{1 - \|\boldsymbol{x}\|^2}\right)^2 g_e, \quad d_p(\boldsymbol{x}, \boldsymbol{y}) = \operatorname{arcosh}\left(1 + 2\frac{\|\boldsymbol{x} - \boldsymbol{y}\|^2}{(1 - \|\boldsymbol{x}\|^2)(1 - \|\boldsymbol{y}\|^2)}\right),$$

where $g_e$ is the Euclidean metric, due to its conformality (angles measured at a point are the same as they are in the actual hyperbolic space), its convenient parameterization, and clear visualization results, the Poincaré ball model is widely used in many applications. However, it can be seen from this equation that the distance within the Poincaré ball model changes rapidly when the points are close to the boundary (i.e. $\|x\| \approx 1$), and hence it is very poorly conditioned.

**Lorentz hyperboloid model.** The Lorentz model is arguably the most natural model algebraically for hyperbolic space. It is defined in terms of a nonstandard scalar product called the *Lorentzian scalar product*. For two-dimensional hyperbolic space, it is defined as

$$\langle \boldsymbol{x}, \boldsymbol{y} \rangle_L = \boldsymbol{x}^T g_l \boldsymbol{y}, \quad \text{where} \quad g_l = \begin{bmatrix} -1 & 0 & 0 \\ 0 & 1 & 0 \\ 0 & 0 & 1 \end{bmatrix}.$$

The Lorentz model of 2-dimensional hyperbolic space is then defined as the Riemannian manifold $(\mathcal{L}^2, g_l)$, where $\mathcal{L}^2$ and associated distance function are given as

$$\mathcal{L}^2 = \{\boldsymbol{x} \in \mathbb{R}^3 : \langle \boldsymbol{x}, \boldsymbol{x} \rangle_L = -1, x_0 > 0\}, \quad d_l(\boldsymbol{x}, \boldsymbol{y}) = \operatorname{arcosh}(-\langle \boldsymbol{x}, \boldsymbol{y} \rangle_L).$$

This model generalizes easily to higher dimensional spaces by increasing the number of 1s on the diagonal of the matrix $g_l$. Points in the Lorentz model lie on the upper sheet of a two-sheeted $n$-dimensional hyperbola. Unlike the Poincaré disk model, which is confined in the Euclidean unit ball, the Lorentz model is unbounded. However, like other models, it can experience severe numerical error for points far away in hyperbolic distance from the origin as shown in Theorem 1.

**Definition 1.** *[Representation error] We are concerned with representing points in hyperbolic space $\mathbb{H}^n$ using floating-points* fl. *Define the* representation error *of a particular point $x \in \mathbb{H}^n$ as $\delta_{\text{fl}}(x) = d_{\mathbb{H}^n}(x, \text{fl}(x))$, and the* worst case representation error *of floating-points representation as a function of the distance-to-origin $d$, which is the maximum representation error of any point with a distance-to-origin at most $d$,*

$$\delta_{\text{fl}}^d = \max_{x \in \mathbb{H}^n, \, d_{\mathbb{H}^n}(x,O) \leq d} \delta_{\text{fl}}(x).$$

**Theorem 1.** *The worst-case representation error (Definition 1) in the Lorentz model using floating-point arithmetic (with machine epsilon $\epsilon_m$) is $\delta_l^d = \text{arcosh}(1 + \epsilon_m(2\cosh^2(d) - 1))$, where $d$ is the hyperbolic distance to origin. This becomes $\delta_l^d = 2d + \log(\epsilon_m) + o(\epsilon_m^{-1}\exp(-2d))$ if $d = O(-\log \epsilon_m)$.*

**Poincaré half-space model.** The Poincaré upper half-space model of the hyperbolic space is the manifold $(\mathcal{U}^n, g_u)$, where $\mathcal{U}^n = \{\boldsymbol{x} \in \mathbb{R}^n : x_n > 0\}$ is the upper half space of the $n$-dimensional Euclidean space. The metric and corresponding distance function is

$$g_u(\boldsymbol{x}) = \frac{g_e}{x_n^2}, \quad d_u(\boldsymbol{x}, \boldsymbol{y}) = \text{arcosh}\left(1 + \frac{\|\boldsymbol{x} - \boldsymbol{y}\|^2}{2x_n y_n}\right)$$

Here $g_e$ is the Euclidean metric. The half-space model is also unbounded and conformal, and has a particularly nice interpretation in two dimensions as a mapping on the complex plane. Note that although it is unbounded, this model still has an "edge" where $x_n = 0$ and it can exhibit numerical issues similar to the Poincaré ball as $x_n$ approaches $0$.

## 4 A Tiling-Based Model for Hyperbolic Plane

As we saw in the previous section, the standard models of hyperbolic space exhibit unbounded numerical error as the hyperbolic distance from the origin increases. In this section, we will describe a tiling-based model that avoids this problem. Our model is constructed on top of the Lorentz model for the two-dimensional hyperbolic plane $\mathbb{H}^2$.

In hyperbolic geometry, a uniform tiling [9, 12, 33] is an edge-to-edge filling of the hyperbolic plane which has regular congruent polygons as faces and is vertex-transitive (there is an isometry mapping any vertex onto any other) [28]. Any tiling is associated with a discrete group $G$ of orientation-preserving isometries of $\mathbb{H}^2$ that preserve the tiling [38, 22]; discrete subgroups of isometries of $\mathbb{H}^2$ (like $G$) are called *Fuchsian groups* [21, 2, 37]. Importantly, not only does the tiling determine $G$, but $G$ also determines the shape of the tiling. One way to see this is to consider the images of a single point in $\mathbb{H}^2$ under the group action $G$ (called an *orbit* of the action). Then the Voronoi diagram associated with the orbit (which partitions each point in $\mathbb{H}^2$ into tiles based on which point in the orbit it is closest to) will be a regular tiling of $\mathbb{H}^2$. This equivalence between tilings and groups means that we can reason about tilings by reasoning about Fuchsian groups.

In the 2-dimensional Lorentz model, isometries can be represented as matrices operating on $\mathbb{R}^3$ that preserve the Lorentzian scalar product. That is, a matrix $A \in \mathbb{R}^{3 \times 3}$ is an isometry if $A^T g_l A = g_l$. If we have some discrete group of isometries $G$, and we choose the tile which contains the origin to be the *fundamental domain* [37, 36] $F$, then we can start to define a tiling-based model on top of the Lorentz model of the hyperbolic plane.

**$L$-tiling model.** Our first insight is to represent points in the hyperbolic plane as a pair consisting of an element of the group and an element of the fundamental domain. The point represented by this pair is the result of the group element applied to the fundamental domain element. For example, the ordered pair $(\boldsymbol{g}, \boldsymbol{x}) \in G \times F$ would represent the point $\boldsymbol{g}\boldsymbol{x}$. The $L$-tiling model of the hyperbolic plane is defined as the Riemannian manifold $(\mathcal{T}_l^n, g_{lt})$, where $g_{lt} = g_l$ and

$$\mathcal{T}_l^n = \{(\boldsymbol{g}, \boldsymbol{x}) \in G \times F : \langle \boldsymbol{x}, \boldsymbol{x} \rangle_L = -1\}, \quad d_{lt}((\boldsymbol{g}_x, \boldsymbol{x}), (\boldsymbol{g}_y, \boldsymbol{y})) = \text{arcosh}\left(-\boldsymbol{x}^T \boldsymbol{g}_x^T g_{lt} \boldsymbol{g}_y \boldsymbol{y}\right).$$

Of course, this is useless unless we have a group $G$ that we can store and compute with easily. Our second insight is to construct a Fuchsian group that can be represented with *integers* so that group operations can be computed exactly and efficiently. The naive way to do this is to try the subgroup of orientation-preserving isometries in $\mathbb{R}^{3 \times 3}$ that have all-integer coordinates: unfortunately, this group (called the modular group) results in a tiling with unbounded fundamental domain, which makes it impossible to bound the representation error, so it is not suitable for our purpose. Instead, we constructed a special Fuchsian group to get a particularly useful $L$-tiling model of hyperbolic plane.

**Definition 2.** *Let $g_a$ and $g_b \in \mathbb{Z}^{3 \times 3}$ and $L \in \mathbb{R}^{3 \times 3}$ be defined as*

$$g_a = \begin{bmatrix} 2 & 1 & 0 \\ 0 & 0 & -1 \\ 3 & 2 & 0 \end{bmatrix}, \quad g_b = \begin{bmatrix} 2 & -1 & 0 \\ 0 & 0 & -1 \\ -3 & 2 & 0 \end{bmatrix}, \quad \text{and } L = \begin{bmatrix} \sqrt{3} & 0 & 0 \\ 0 & 1 & 0 \\ 0 & 0 & 1 \end{bmatrix}.$$

*Define $G$ to be the fuchsian group generated by $L \cdot g_a \cdot L^{-1}$ and $L \cdot g_b \cdot L^{-1}$. It is straightforward to verify that $(L \cdot g_a \cdot L^{-1})^T g_l (L \cdot g_a \cdot L^{-1}) = (L \cdot g_b \cdot L^{-1})^T g_l (L \cdot g_b \cdot L^{-1}) = g_l$. Note that*

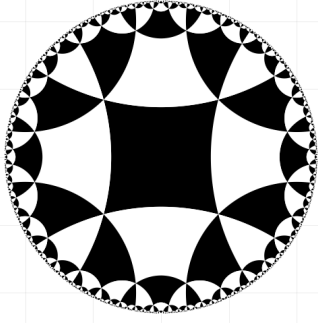

Figure 1: The regular quadrilateral tiling of hyperbolic space produced by the group $G$ on the Poincaré disk.

---
**Algorithm 1** Map Lorentz model to $L$-tiling model

---
**Require:** $x \in \mathcal{L}^2$
    **initialize** $R \leftarrow I$
    **while** $x \notin F$ **do**
        **if** $x_2 \leq -|x_3|$ **then** $S \leftarrow g_a^{-1}$
        **else if** $x_2 \geq |x_3|$ **then** $S \leftarrow g_b^{-1}$
        **else if** $x_3 < -|x_2||$ **then** $S \leftarrow g_b$
        **else if** $x_3 > |x_2|$ **then** $S \leftarrow g_a$
        $(R, x) \leftarrow (R \cdot S, L \cdot S^{-1} \cdot L^{-1} \cdot x)$
        $x_1 = \sqrt{x_2^2 + x_3^2 + 1}$    ▷ renormalize $x$
    **end while**
**output** $(R, x)$

---

$g_a^6 = g_b^6 = (g_a g_b)^3 = I$, and so this group has presentation

$$G = L \cdot \langle g_a, g_b | g_a^6 = g_b^6 = (g_a g_b)^3 = 1 \rangle \cdot L^{-1}.$$

Importantly, *any* element of $G$ can be represented in the form $g = LZL^{-1}$ where $Z \in \mathbb{Z}^{3\times 3}$ is an all-integers matrix. For this reason, we can store elements of $G$ and take group products and inverses using *only integer arithmetic*. This property makes $G$ of particular interest for use with an $L$-tiling model. But before we can construct an $L$-tiling model for this group, we need to choose an appropriate fundamental domain.

**Theorem 2.** $F = \{(x_1, x_2, x_3) \in \mathcal{L}^2 | \max(2x_2^2 - x_3^2, 2x_3^2 - x_2^2) < 1\}$ *is a fundamental domain of* $G$. *Any point in $\mathcal{L}^2$ can be mapped by $G$ to one unique point in $F$ or to a point on its boundary.*

Figure 1 illustrates the tiling generated by group $G$ and $F$ centered at the origin in the Poincaré disk model. Now that we have a group and a fundamental domain, we can start computing with our new $L$-tiling model. The first step is to build a relationship between standard hyperbolic models and the $L$-tiling model, i.e., convert points into the $L$-tiling model from other models: to this end, we offer a "normalization" procedure (Algorithm 1), which transforms the Lorentz model to the $L$-tiling model. The convergence and complexity of this algorithm are characterized in Theorem 3.

**Theorem 3.** *For any point in the Lorentz model, Algorithm 1 converges and stops within $1 + 7d$ steps, where $d = d(x, O)$ denotes the distance from $x$ to the origin.*

**Representing points.** For a point $(g, x)$ in the $L$-tiling model, where $g \in G$, $x \in F$, we represent this point with $(g, \mathrm{fl}(x))$. Here $g$ is exact because it is represented by the related integer matrix, while fl denotes float arithmetic with error bounded by some machine epsilon $\epsilon_m$. This floating point arithmetic introduces some representation error, which we can bound as follows:

$$d_{lt}((g, x), (g, \mathrm{fl}(x))) = \operatorname{arcosh}(-x^T g^T g_{lt} g \mathrm{fl}(x)) = \operatorname{arcosh}(-x^T g_{lt} \mathrm{fl}(x))$$

Since $x \in F$, which is bounded as shown in Theorem 2, this approximation error can also be bounded (Theorem 4). In comparison, for the Lorentz model, the worst case error (Theorem 1) is unbounded.

**Theorem 4.** *The representation error (Definition 1) in $L$-tiling model is bounded as $\delta_{lt}^d \leq \sqrt{5\epsilon_m} + 15\epsilon_m/4 + o(\epsilon_m)$, where $\epsilon_m$ is the machine error.*

By convention, for $(g, x)$ in the $L$-tiling model, where $g \in G$, $x \in F$, firstly we will usually denote $g$ using its related integer matrix $\hat{g} = L^{-1}gL$; Secondly for the point $x \in F$, even though $x$ is part of the Lorentz model and lies in 3-dimensional space, in fact only two coordinates suffice to determine its position. For simplicity, we define a bijective function $h(x_2, x_3) = (\sqrt{1 + x_2^2 + x_3^2}, x_2, x_3)$ which maps $\mathbb{R}^2$ to the hyperboloid model (this is sometimes called the *Gans model* [17]). In this way, we can represent $(g, x) \in \mathcal{T}_{lt}^2$ as $(\hat{g}, h^{-1}(x))$. We can then store the integer matrix and floating-point coordinates $h^{-1}(x) \in \mathbb{R}^2$. In future sections, we assume we will use this integer matrix and two-coordinate representation rather than $(g, x)$ unless otherwise specified.

# 5 Learning in the $L$-tiling Model

In this section, we provide an efficient and precise way to compute distances and gradients accordingly in the $L$-tiling model, with which we can construct learning algorithms to train and derive embeddings. We also present error bounds for these computations, which avoid the "Nan" problem.

**Distance and Gradient.** For two points $(U, u), (V, v)$ in the $L$-tiling model, the formula to compute distance is

$$d((U, u), (V, v)) = \text{arcosh}(h(u)^T L^{-T} Q L^{-1} h(v))$$

where $Q = -U^T L^T g_{lt} L V$ can be computed exactly with integer arithmetic. A potential difficulty here is that the entries in $Q$ can be very large (possibly even larger than can be represented in floating-point). To solve this, observe that $Q_{11}$ has the largest absolute value in the matrix (Lemma 2). So we define and compute $\hat{Q} = Q/Q_{11}$, which is guaranteed to not overflow the floating-point format, since all the entries of $\hat{Q}$ are in $[-1, 1]$. Let $d_c = h(u)^T L^{-T} \hat{Q} L^{-1} h(v)$, this reduces our distance to

**Algorithm 2** RSGD in the $L$-tiling model

---
**Require:** Objective function $f$, fuchsian group $G$ with fundamental domain $F$, exponential map $\exp_{\beta_t}(v) = \cosh(\|v\|_L)\beta_t + \sinh(\|v\|_L)\frac{v}{\|v\|_L}$, where $\|v\|_L = \sqrt{\langle v, v \rangle_L}$.
**Require:** $(\beta_t, U_t) \in F \times G$, Epochs $T$, and learning rate $\eta$
  **for** $t = 0$ to $T - 1$ **do**
    $l_t \Leftarrow g_{\beta_t}^{-1} \nabla_{\beta_t} f(LU_t L^{-1} \beta_t)$ ▷ Riemannian
    grad $f \Leftarrow l_t + \langle \beta_t, l_t \rangle_L \beta_t$   ▷ Projection
    $\beta_{t+1} \Leftarrow \exp_{\beta_t}(-\eta \text{ grad} f)$  ▷ Update
    **if** $\beta_{t+1} \notin F$ **then**
      $W \Leftarrow \arg \min_{W \in G} d(LW^{-1}L^{-1}\beta_{t+1}, O)$
      $U_{t+1} \Leftarrow U_t \cdot W$  ▷ Normalize if $\beta_{t+1} \notin F$
      $\beta_{t+1} \Leftarrow LW^{-1}L^{-1}\beta_{t+1}$
    **else**
      $U_{t+1} \Leftarrow U_t$
    **end if**
  **end for**
**output** $(\beta_{t+1}, U_{t+1})$

---

$$d((U, u), (V, v)) = \text{arcosh}(Q_{11} \cdot d_c) = \log(Q_{11}) + \log\left(d_c + \sqrt{d_c^2 - Q_{11}^{-2}}\right)$$

Note that (assuming that we can compute $\log(Q_{11})$ without overflow) this expression can be computed in floating-point without any overflow, since all the numbers involved are well within range. The corresponding formula for the gradient can also be derived as

$$\nabla_u d((U, u), (V, v)) = \frac{\nabla h(u)^T L^{-T} \hat{Q} L^{-1} h(v)}{\sqrt{d_c^2 - Q_{11}^{-2}}}, \quad \text{where } \nabla h(u) = \left[\frac{u}{\sqrt{1 + \|u\|^2}}, \ I\right].$$

Again, this avoids any possibility of overflow. We provide the error of computing distance (Theorem 6) and gradient (Theorem 7) in $L$-tiling model together with that in Lorentz model in Appendix. By computing with integer arithmetic, the error will be independent of how far the points are from the origin, which guarantees that it avoids the "NaN" problem. Since we can compute distances and derivatives, we can use all the standard gradient-based optimization algorithms. In Algorithm 2 we present the most powerful one, RSGD, adapted for use with the $L$-tiling model.

# 6 Extension to Higher Dimensional Space

Extending the $L$-tiling model to higher dimension seems simple: just find a cocompact (to ensure a bounded fundamental domain) discrete subgroup of the higher-dimensional space's isometry group. Such a group would induce a *honeycomb*, a higher-dimensional analog of a regular tiling of the hyperbolic plane. Unfortunately, a classic result by Coxeter [10] says this is impossible in general: there are no such regular honeycombs in six or more dimensions.

In order to derive a high dimensional tiling-based model which may be necessary for complicated datasets, we consider two possibilities.

- Take the Cartesian product of multiple copies of the $L$-tiling model in the hyperbolic plane. The use of multiple copies of models in the hyperbolic plane was previously proposed in Gu et al. [19].
- Construct honeycombs and tilings from a set of isometries that is not a group.

Practically we can embed data into products of $\mathbb{H}^2$s as we do in Section 7, however, the first possibility (tilings over $\mathbb{H}^2 \times \mathbb{H}^2 \times \cdots \mathbb{H}^2$) is something fundamentally different from tiling a single

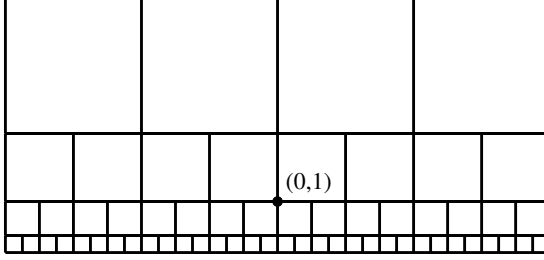

| Datasets | Nodes | Edges |
|---|---|---|
| Bio-yeast[29] | 1458 | 1948 |
| WordNet[14] | 74374 | 75834 |
| ↳ Nouns | 82115 | 769130 |
| ↳ Verbs | 13542 | 35079 |
| ↳ Mammals | 1181 | 6541 |
| Gr-QC[23] | 4158 | 13422 |

Figure 2: (Left) The infinite square tiling of hyperbolic space on the half-plane model; (Right) Datasets.

high dimensional hyperbolic space (tilings over $\mathbb{H}^n$), which we aims to do in this section. Fortunately for the second possibility, in half-space model, we find that horizontal translation and homotheties are hyperbolic isometries, which can produce the (infinite) square tiling illustrated in Figure 2 [1, 6]. It consists of the image of the unit square $S$, with vertical and horizontal sides and whose lower left corner is at $(0, ..., 0, 1)$, under the maps

$$p \to 2^j(p+k), (j, k) \in \mathbb{Z} \times (\mathbb{Z}^{n-1} \times \{0\}).$$

Here each square is isometric to every other square, and the unit square $S$ takes on the role of the fundamental domain in Theorem 2. With these maps, we can define a tiling-based model on top of the half-space model as follows.

$H$**-tiling model.** The $H$-tiling model of the hyperbolic space is defined as the Riemannian manifold $(\mathcal{T}_h^n, g_{ht})$, where

$$\mathcal{T}_h^n = \{(j, \boldsymbol{k}, \boldsymbol{x}) \in \mathbb{Z} \times (\mathbb{Z}^{n-1} \times \{0\}) \times S\}, \qquad g_{ht}(j, \boldsymbol{k}, \boldsymbol{x}) = \frac{g_e}{(2^j x_n)^2}$$

The associated distance function on $\mathcal{T}_h^n$ is then given as

$$d((j_1, \boldsymbol{k}_1, \boldsymbol{x}), (j_2, \boldsymbol{k}_2, \boldsymbol{y})) = \operatorname{arcosh}\left(1 + \frac{\|2^{j_1} z_1 - 2^{j_2} z_2 + 2^{j_1} k_1 - 2^{j_2} k_2\|^2}{2^{j_1+j_2+1} z_{1n} z_{2n}}\right).$$

Similarly, we derive the representation error for this model, which is bounded by a constant depending on the machine epsilon as shown in Theorem 5.

**Theorem 5.** *The representation error (Definition 1) in $H$-tiling model is bounded as $\delta_{ht}^d = \sqrt{(n+3)\epsilon_m}/2 + (n+3)\epsilon_m/4 + o(\epsilon_m)$, where $\epsilon_m$ is the machine error.*

We can compute distances and gradients in a numerically accurate way, and run RSGD algorithm on this model for optimization, just as we could in the $L$-tiling model. For lack of space, we defer that discussion and more learning details of Sections 5 and 6 to Appendix A. Also note that we are not tied to the half-space model here: while the half-space model gives a convenient way to describe the set of transformations we are using, we could use the same transformations with any underlying model we choose by adding an appropriate conversion.

## 7 Experiments

**Compressing embeddings.** We consider storing 2-dimensional embeddings using the $L$-tiling model for compression: storage using few bits. While storing the integer matrices exactly is convenient for computation, it does tend to take up a lot of extra memory (especially when BigInts are needed to store the integer values in the matrix). This motivates us to look for alternative storage methods. To store the matrix $g$, we prropose and evaluate the following methods:

- Matrix: store all 9 integers in the matrix $g$ as Int or BigInt.
- Entries: store just $g_{21}, g_{31}$ as Int or BigInt, which we can show is sufficient to reconstruct the whole matrix (Lemma 1 in Appendix D).
- Order: store the generator order with respect to $g_a, g_b$ as a string.
- VBW: store the generator order with respect to $g_a, g_b$ using a variable bit-width encoding. We use binary code 10 to represent $g_a^1$ and $g_b^1$, 001 to represent $g_a^2$ and $g_b^2$, 010 to represent $g_a^3$ and $g_b^3$, 011 to represent $g_a^4$ and $g_b^4$, 11 to represent $g_a^5$ and $g_b^5$, and 000 to represent the end of the string. This encoding disambiguates the generators by taking advantage of the fact that powers of $g_a$ and $g_b$ must alternate to appear.

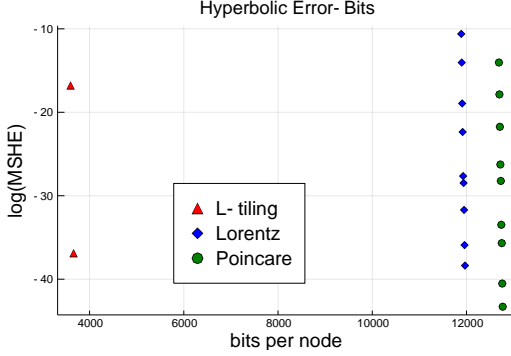

| Models | size (MB) | bzip (MB) |
|---|---|---|
| Poincaré(16512B) | 372 | 119 |
| Poincaré(12688B) | 287 | 81 |
| Lorentz(11898B) | 396 | 171 |
| Matrix(VLQ) | 600(286) | 260(251) |
| Entries(VLQ) | 132(63) | 57(55) |
| Order | 111 | 8.52 |
| VBW | 33.1 | 6.07 |
| fpt-f32 | 6.2 | 1.96 |
| fpt-f16 | 4.25 | 1.07 |

Figure 3: (Left) Hyperbolic error for WordNet Nouns; (Right) Compression statistics for WordNet under the same MSHE, first block contains the size of original poincare embedding, second block contains the size of compressed baseline models, third block contains the size of matrix part in the $L$-tiling model (size of compressed integers using VLQ is also reported), the last block contains size of float points (fpt, f32 or f16) in the fundamental domain of $L$-tiling model.

The generator order and corresponding VBW encoding of a given matrix can be derived using Algorithm 1 as shown in Lemma 1. Additionally, for Int or BigInt, we can use variable length quantity (VLQ) to compress [31]. To test our compression methods, we use combinatorial construction [30] to derive 2-dimensional Poincaré disk embeddings for WordNet (Tree-like) and Bio-yeast datasets (Figure 2), then we transform embeddings and compress them. We calculate the mean squared hyperbolic error (MSHE) with respect to the original embedding to show the error of compression.

For Bio-yeast, we evaluate different compressions using MSHE and mean average precision (MAP). As shown in Table 1, representation and compression in the $L$-tiling model (with different floating number for points in the fundamental domain) does not hurt MAP performance, while the compression of the Poincaré embedding to the same size hurts MAP severely. For WordNet, we plot the scatter of the relationship between $\log(\text{MSHE})$ and bits to store per node in Figure 3. Under the same MSHE, the $L$-tiling model requires approximately $2/3$ less bits per node compared to that of Lorentz and Poincaré models. We measure the size of different models under the same MSHE in Figure 3. The $L$-tiling model can represent the hyperbolic embedding with only $(6.07+1.07)$ MB, which is $2\%$ of the original 372 MB, while it will cost at least 81 MB for any reasonably accurate baseline model.

**Learning embeddings.** As we have shown, our tiling-based models represent hyperbolic space accurately, and so they can be used for learning embeddings with generic objective functions. However, since we analyzed hyperbolic distance and gradient computation error in this paper, we evaluate our learning methods empirically on objective functions that depend on distances. As proposed by Nickel and Kiela [26], to consider the ability to embed data that exhibits a clear latent hierarchical structure, we conduct reconstruction experiments on the transitive closure of the Gr-QC, WordNet Nouns, Verbs and Mammals hierarchy as summarized in Table 2. We firstly embed the data and then reconstruct it from the embedding to evaluate the representation capacity of the embedding. Let $\mathcal{D} = \{(u, v)\}$ be the set of observed relations between objects. We aim to learn embeddings of $\mathcal{D}$ such that related objects are close in the embedding space. To do this, we minimize the loss [26]

$$\mathcal{L}(\Theta) = \sum_{(u,v)\in\mathcal{D}} \log \frac{e^{-d(\boldsymbol{u},\boldsymbol{v})}}{\sum_{\boldsymbol{v}'\in\mathcal{N}(u)} e^{-d(\boldsymbol{u},\boldsymbol{v}')}}, \tag{1}$$

where $\mathcal{N}(u) = \{v \mid (u,v) \notin \mathcal{D}\} \cup \{u\}$ is the set of negative examples for $u$ (including $u$). We randomly sample $|\mathcal{N}(u)| = 50$ negative examples per positive example during training.

Table 1: Compression of Bio-yeast

| Models | MSHE | MAP |
|---|---|---|
| Poincaré(8128B) | 0.00 | 0.873 |
| Poincaré(6360B) | 4.84e-17 | 0.873 |
| Poincaré(1832B) | 1.01e+03 | 0.310 |
| $L$-tiling-f64(1832B) | 9.76e-17 | 0.873 |
| $L$-tiling-f32(1768B) | 5.12e-08 | 0.873 |
| $L$-tiling-f16(1736B) | 4.30e-05 | 0.873 |
| $L$-tiling-f0(1704B) | 5.90e-01 | 0.873 |

Table 2: Learning Mammals

| Models | MAP | MR |
|---|---|---|
| Poincaré | 0.805±0.011 | 2.22±0.10 |
| Lorentz | 0.855±0.013 | 1.89±0.13 |
| $L$-tiling-SGD | 0.892±0.031 | 2.14±0.70 |
| $L$-tiling-RSGD | **0.930**±0.005 | **1.49**±0.09 |
| $H$-tiling-RSGD | 0.923±0.016 | 1.56±0.20 |

| DIMENSION | MODELS | WORDNET NOUNS | | WORDNET VERBS | | GR-QC | |
|---|---|---|---|---|---|---|---|
| | | MAP | MR | MAP | MR | MAP | MR |
| 2 | POINCARÉ | 0.124±0.001 | 68.75±0.26 | 0.537±0.005 | 4.74±0.17 | 0.561± 0.004 | 67.91±1.14 |
| | LORENTZ | 0.382±0.004 | 17.80±0.55 | **0.750**±0.004 | 2.11±0.06 | 0.563±0.003 | 68.40±1.20 |
| | $H$-TILING-RSGD | 0.390±0.002 | 17.18±0.52 | 0.747±0.003 | 2.10±0.05 | 0.560±0.004 | 66.17±1.05 |
| | $L$-TILING-SGD | 0.341±0.001 | 20.27±0.39 | 0.696±0.003 | 2.33±0.07 | **0.574**±0.005 | **63.04**±1.97 |
| | $L$-TILING-RSGD | **0.413**±0.007 | **15.26**±0.57 | 0.746±0.004 | **2.07**±0.03 | 0.564± 0.002 | 63.88±1.47 |
| 4 | 2×LORENTZ | 0.460±0.001 | 10.12±0.03 | **0.873**±0.001 | 1.31±0.01 | **0.718**±0.003 | 11.59±0.32 |
| | 2×$L$-TILING-RSGD | **0.464**±0.002 | **9.99**±0.09 | 0.871±0.004 | 1.33±0.01 | 0.716±0.005 | **10.88**±0.42 |
| 5 | POINCARÉ | 0.848±0.001 | 4.16±0.04 | 0.948±0.001 | 1.19±0.01 | 0.714±0.000 | 34.60±0.52 |
| | LORENTZ | 0.865±0.005 | **3.70**±0.12 | 0.947±0.001 | **1.16**±0.00 | **0.715**±0.003 | 33.51± 1.04 |
| | $H$-TILING-RSGD | **0.869**±0.001 | **3.70**±0.06 | **0.949**±0.001 | **1.16**±0.01 | 0.714±0.002 | **33.46**±0.66 |
| 10 | POINCARÉ | 0.876±0.001 | 3.47±0.02 | 0.953±0.002 | 1.16±0.01 | 0.729±0.000 | 29.51±0.21 |
| | LORENTZ | 0.865±0.004 | 3.36±0.04 | 0.948±0.001 | 1.15±0.00 | 0.724±0.001 | 29.34±0.23 |
| | $H$-TILING-RSGD | **0.888**±0.004 | **3.22**±0.02 | 0.954±0.002 | 1.15±0.00 | 0.729±0.001 | 27.75±0.39 |
| | 5×LORENTZ | 0.672±0.000 | 4.42±0.00 | 0.958±0.003 | 1.07±0.01 | 0.944±0.007 | 3.06±0.03 |
| | 5×$L$-TILING-RSGD | 0.674±0.000 | 4.41±0.00 | **0.961**±0.002 | **1.06**±0.00 | **0.953**±0.002 | **3.03**±0.01 |

Table 3: Learning experiments on different datasets. Results are averaged over 5 runs and reported in mean+std style.

We consider the $L$-tiling models trained with RSGD and SGD, $H$-tiling models trained with RSGD and the Cartesian product of multiple copies of 2-dimensional $L$-tiling models (proposed in Gu et al. [19]). The Poincaré ball model [26] and Lorentz model [27] were included as baselines. All models were trained in float64 for 1000 epochs with the same hyper-parameters. To evaluate the quality of the embeddings, we make use of the standard graph embedding metrics in [3, 25]. For an observed relationship $(u, v)$, we rank the distance $d(\boldsymbol{u}, \boldsymbol{v})$ among the set $\{d(u, v')|(u, v') \in \mathcal{D}\}$, then we evaluate the ranking on all objects in the dataset and record the mean rank (MR) as well as the mean average precision (MAP) of the ranking.

We start by evaluating all 2-dimensional embeddings on the Mammals dataset. As shown in Table 2, all tiling-based models outperform baseline models: the performances of $L$-tiling model and $H$-tiling model with RSGD are nearly the same. In particular, the $L$-tiling model achieves a $8.8\%$ MAP improvement on Mammals compared to Lorentz model.

Embedding experiments on other three large datasets are presented in Table 3. These results show that tiling-based models generally perform better than baseline models in various dimensions. We found three observations particularly interesting here. First, the group-based tiling model ($L$-tiling) performs better than the non-group tiling model ($H$-tiling) in two dimensions. Second, tiling-based models perform particularly better than baseline models for the largest WordNet Nouns dataset, which further validates that numerical issue happens when the embeddings are far from the origin and affects the embedding performances. Third, the Cartesian product of multiple copies of 2-dimensional $L$-tiling models performs even better than high dimensional models when the datasets are not too large and complex such as WordNet Verbs and Gr-QC, especially for the dense graph Gr-QC.

More experiment details are provided in Appendix B. We release our compression code[*] in Julia and learning code[†] in PyTorch publicly for reproducibility.

# 8 Discussions and Conclusions

In this paper, we introduced tiling-based models of hyperbolic space, which use a tiling backed by integer arithmetic to represent any point in hyperbolic space with fixed and provably bounded error. We showed that $L$-tiling model using one particular group $G$ can achieve substantial compression of an embedding with minimal loss, and can perform well on embedding tasks compared with other methods. A notable observation that could motivate future work is that our group based tiling model ($L$-tiling) performs better than the non-group tiling model ($H$-tiling) in two dimensions: it is interesting to ask if this reflects some advantages of the group, and if we can use this to find better non-regular tilings in high dimensions. Overall, it is our hope that this work can help make hyperbolic embeddings more numerically robust and thereby make them easier for practitioners to use.

---

[*]`https://github.com/ydtydr/HyperbolicTiling_Compression`
[†]`https://github.com/ydtydr/HyperbolicTiling_Learning`

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
