[Supplementary Material]

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

Since $\boldsymbol{x} \in F$, which is bounded as shown in Theorem 2, this approximation error can also be bounded (Theorem 4). In comparison, for the Lorentz model, the worst case error (Theorem 1) is unbounded.

**Theorem 4.** *The representation error (Definition 1) in $L$-tiling model is bounded as $\delta_{lt}^d \leq \sqrt{5\epsilon_m} + 15\epsilon_m/4 + o(\epsilon_m)$, where $\epsilon_m$ is the machine error.*

By convention, for $(g, \boldsymbol{x})$ in the $L$-tiling model, where $g \in G$, $\boldsymbol{x} \in F$, firstly we will usually denote $g$ using its related integer matrix $\hat{g} = L^{-1}gL$; Secondly for the point $x \in F$, even though $x$ is part of the Lorentz model and lies in 3-dimensional space, in fact only two coordinates suffice to determine its position. For simplicity, we define a bijective function $h(x_2, x_3) = (\sqrt{1 + x_2^2 + x_3^2}, x_2, x_3)$ which maps $\mathbb{R}^2$ to the hyperboloid model (this is sometimes called the *Gans model* [17]). In this way, we can represent $(g, \boldsymbol{x}) \in \mathcal{T}_{lt}^2$ as $(\hat{g}, h^{-1}(\boldsymbol{x}))$. We can then store the integer matrix and floating-point coordinates $h^{-1}(\boldsymbol{x}) \in \mathbb{R}^2$. In future sections, we assume we will use this integer matrix and two-coordinate representation rather than $(g, \boldsymbol{x})$ unless otherwise specified.

# 5    Learning in the $L$-tiling Model

In this section, we provide an efficient and precise way to compute distances and gradients accordingly in the $L$-tiling model, with which we can construct learning algorithms to train and derive embeddings. We also present error bounds for these computations, which avoid the "Nan" problem.

**Distance and Gradient.**    For two points $(U, u), (V, v)$ in the $L$-tiling model, the formula to compute distance is

$$d((U, u),(V, v)) = \text{arcosh}(h(u)^T L^{-T} Q L^{-1} h(v))$$

where $Q = -U^T L^T g_{lt} L V$ can be computed exactly with integer arithmetic. A potential difficulty here is that the entries in $Q$ can be very large (possibly even larger than can be represented in floating-point). To solve this, observe that $Q_{11}$ has the largest absolute value in the matrix (Lemma 2). So we define and compute $\hat{Q} = Q/Q_{11}$, which is guaranteed to not overflow the floating-point format, since all the entries of $\hat{Q}$ are in $[-1, 1]$. Let $d_c = h(u)^T L^{-T} \hat{Q} L^{-1} h(v)$, this reduces our distance to

**Algorithm 2** RSGD in the $L$-tiling model

**Require:** Objective function $f$, fuchsian group $G$ with fundamental domain $F$, exponential map $\exp_{\beta_t}(v) = \cosh(\|v\|_L)\beta_t + \sinh(\|v\|_L)\frac{v}{\|v\|_L}$, where $\|v\|_L = \sqrt{\langle v, v \rangle_L}$.
**Require:** $(\beta_t, U_t) \in F \times G$, Epochs $T$, and learning rate $\eta$
$\quad$**for** $t = 0$ to $T - 1$ **do**
$\quad\quad l_t \Leftarrow g_{\beta_t}^{-1} \nabla_{\beta_t} f(LU_t L^{-1}\beta_t)$ ▷ Riemannian
$\quad\quad \text{grad} f \Leftarrow l_t + \langle \beta_t, l_t \rangle_L \beta_t$ $\quad$ ▷ Projection
$\quad\quad \beta_{t+1} \Leftarrow \exp_{\beta_t}(-\eta \text{ grad} f)$ $\quad$ ▷ Update
$\quad\quad$**if** $\beta_{t+1} \notin F$ **then**
$\quad\quad\quad W \Leftarrow \arg\min_{W \in G} d(LW^{-1}L^{-1}\beta_{t+1}, O)$
$\quad\quad\quad U_{t+1} \Leftarrow U_t \cdot W$ ▷ Normalize if $\beta_{t+1} \notin F$
$\quad\quad\quad \beta_{t+1} \Leftarrow LW^{-1}L^{-1}\beta_{t+1}$
$\quad\quad$**else**
$\quad\quad\quad U_{t+1} \Leftarrow U_t$
$\quad\quad$**end if**

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

$$
\begin{cases}
\nabla_u d((U, u), (V, v)) = \frac{\nabla h(u)^T L^{-T} \hat{Q} L^{-1} h(v)}{\sqrt{(h(u)^T L^{-T} \hat{Q} L^{-1} h(v))^2 - Q_{11}^{-2}}} = \frac{\nabla h(u)^T L^{-T} \hat{Q} L^{-1} h(v)}{\sqrt{d_c^2 - Q_{11}^{-2}}} \\
\nabla_v d((U, u), (V, v)) = \frac{\nabla h(v)^T L^{-T} \hat{Q}^T L^{-1} h(u)}{\sqrt{(h(u)^T L^{-T} \hat{Q} L^{-1} h(v))^2 - Q_{11}^{-2}}} = \frac{\nabla h(v)^T L^{-T} \hat{Q}^T L^{-1} h(u)}{\sqrt{d_c^2 - Q_{11}^{-2}}}
\end{cases}
$$

where

$$
\nabla h(u) = \left[\frac{u}{\sqrt{1 + ||u||^2}}, \ I\right], \nabla h(v) = \left[\frac{v}{\sqrt{1 + ||v||^2}}, \ I\right]
$$

We provide the error bound for distance and gradient computation in $L$-tiling model using float arithmetic in Theorem 6, Theorem 7, where errors are independent of how far points are from the origin and solves the "Nan" problem.

**SGD in the $L$-tiling model**   We offer SGD algorithm below, with the addition of a normalization when the parameter goes out of the $L$-tiling model, which is performed using Algorithm 1, whose convergence and complexity were shown in Theorem 3.

---

**Algorithm 3** SGD using group representation

---

**Require:** Objective function $f$, fuchsian group $G$ with fundamental domain $F \subset \mathbb{R}^2$
**Require:** Tuple $(\theta_t, U_t) \in F \times G$
**Require:** Number of epochs $T$, and learning rate $\alpha$
    **for** $t =$ to $T - 1$ **do**
        $l_t \Leftarrow \nabla_{\theta_t} f(L U_t L^{-1} h(\theta_t))$              ▷ Euclidean gradient w.r.t. $\theta_t$
        $\theta_{t+1} \Leftarrow \theta_t - \alpha l_t$                    ▷ Update $\theta_t \in F$
        **if** $\theta_{t+1} \notin F$ **then**
            $W \Leftarrow \arg\min_{W \in G} d(L W^{-1} L^{-1} h(\theta_{t+1}), 0)$
            $U_{t+1} \Leftarrow U_t \cdot W$                ▷ Normalize if $\theta_{t+1} \notin F$
            $\theta_{t+1} \Leftarrow L W^{-1} L^{-1} h(\theta_{t+1})$
        **else**
            $U_{t+1} \Leftarrow U_t$
        **end if**
    **end for**
**output** $(\theta_{t+1}, U_{t+1})$

---

**RSGD in the $L$-tiling model**   We show RSGD in the $L$-tiling model here, which is in correspondence to Algorithm 1. Equivalence of this algorithm to that in Lorentz model is shown in Theorem D For $(U_t, u_t)$ in the $L$-tiling model, let $f$ be the objective function, denote $\nabla_{u_t} f$ to be the Euclidean

gradient of $f$ w.r.t. $u_t$. To do RSGD in the $L$-tiling model, firstly transform the Euclidean gradient to Riemannian gradient using the pull-back hyperbolic metric:

$$h_t = g_{u_t}^{-1} \nabla_{u_t} f,$$

then project it into the tangent space at $u_t$,

$$\text{grad}_{u_t} f = h_t + \langle u_t, h_t \rangle_L u_t.$$

then the RSGD algorithm in the $L$-tiling model updates $u^t$ as follows:

$$u_{t+1} = \exp_{u_t}(v) = \cosh\left(||v||_L\right) u_t + \sinh\left(||v||_L\right) \frac{v}{||v||_L},$$

where $v = -\eta \cdot \text{grad}_{u_t} f$ and $||v||_L = \sqrt{\langle v, v \rangle_L}$.

**Efficient Computation in $H$-tiling Model**   For points $(j_1, \boldsymbol{k}_1, \boldsymbol{z}_1)$ and $(j_2, \boldsymbol{k}_2, \boldsymbol{z}_2)$ in $H$-tiling model, directly computing distance as follows works in most cases,

$$d_h((j_1, \boldsymbol{k}_1, \boldsymbol{z}_1), (j_2, \boldsymbol{k}_2, \boldsymbol{z}_2)) = \text{arcosh}(1 + \frac{||2^{j_1}(z_1 + k_1) - 2^{j_2}(z_2 + k_2)||^2}{2^{j_1 + j_2 + 1} z_{1n} z_{2n}})$$

However, we do consider situations where overflows may happen and provide an alternative way to compute distance. Suppose without loss of generality that $j_2 \geq j_2$, then we can write distance as

$$d_h((j_1, \boldsymbol{k}_1, \boldsymbol{z}_1), (j_2, \boldsymbol{k}_2, \boldsymbol{z}_2)) = \text{arcosh}(1 + 2^{j_1 - j_2} \frac{||z_1 - 2^{j_2 - j_1} z_2 + k_1 - 2^{j_2 - j_1} k_2||^2}{2 z_{1n} z_{2n}})$$

let $k_1 - 2^{j_2 - j_1} k_2 = 2^s I$ where $s$ is some natural number scale factor such that $||I|| < 1$. This is easy to compute exactly using integer arithmetic. (Note that $k_1 - 2^{j_2 - j_1} k_2$ is an integer vector, choose $s = \lceil \log_2(||k_1 - 2^{j_2 - j_1} k_2||^2)/2 \rceil$, where the values inside the $\log_2$ are all integers). Then we get

$$d_h((j_1, \boldsymbol{k}_1, \boldsymbol{z}_1), (j_2, \boldsymbol{k}_2, \boldsymbol{z}_2)) = \text{arcosh}(1 + 2^{j_1 - j_2} \frac{||2^s I + z_1 - 2^{j_2 - j_1} z_2||^2}{2 z_{1n} z_{2n}})$$

$$= \text{arcosh}(1 + 2^{2s + j_1 - j_2} \frac{||I + 2^{-s} z_1 - 2^{j_2 - j_1 - s} z_2||^2}{2 z_{1n} z_{2n}})$$

$$= \text{arcosh}(1 + 2^{2s + j_1 - j_2} X)$$

$$= \log(1 + 2^{2s + j_1 - j_2} X + \sqrt{(1 + 2^{2s + j_1 - j_2} X)^2 - 1})$$

$$= (2s + j_1 - j_2)\log(2) + \log(2^{-2s - j_1 + j_2} + X + \sqrt{X^2 + 2^{1 - 2s - j_1 + j_2} X})$$

where

$$X = \frac{||I + 2^{-s} z_1 - 2^{j_2 - j_1 - s} z_2||^2}{2 z_{1n} z_{2n}}$$

then we can get following gradients,

$$\begin{cases} \dfrac{\partial d}{\partial X} = \dfrac{1 + \dfrac{X + 2^{-2s - j_1 + j_2}}{\sqrt{X^2 + 2^{1 - 2s - j_1 + j_2} X}}}{2^{-2s - j_1 + j_2} + X + \sqrt{X^2 + 2^{1 - 2s - j_1 + j_2} X}} \\[3mm] \nabla_{z_{1i}} X = \dfrac{I_i + 2^{-s} z_{1i} - 2^{j_2 - j_1 - s} z_{2i}}{2^s z_{1n} z_{2n}} \\[3mm] \nabla_{z_{2i}} X = -\dfrac{I_i + 2^{-s} z_{1i} - 2^{j_2 - j_1 - s} z_{2i}}{2^{s + j_1 - j_2} z_{1n} z_{2n}} \\[3mm] \nabla_{z_{1n}} X = \dfrac{I_n + 2^{-s} z_{1n} - 2^{j_2 - j_1 - s} z_{2n}}{2^s z_{1n} z_{2n}} - \dfrac{||I + 2^{-s} z_1 - 2^{j_2 - j_1 - s} z_2||^2}{2 z_{1n}^2 z_{2n}} \\[3mm] \nabla_{z_{2n}} X = -\dfrac{I_n + 2^{-s} z_{1n} - 2^{j_2 - j_1 - s} z_{2n}}{2^{s + j_1 - j_2} z_{1n} z_{2n}} - \dfrac{||I + 2^{-s} z_1 - 2^{j_2 - j_1 - s} z_2||^2}{2 z_{1n} z_{2n}^2} \end{cases}$$

Using chain rule, Euclidean gradients of $d_h$ w.r.t. $z_1, z_2$ can be derived. We provide the error bound for this distance computation in $H$-tiling model using float arithmetic in Theorem 8. we neglect the error bound for gradient computation here which can also be derived similarly. In $H$-tiling model,

these computation errors are independent of how far points are from the origin and solves the "Nan" problem.

**RSGD in the $H$-tiling model** For $(j_t, k_t, z^t)$ in the $H$-tiling model, let $f$ be the objective function, denote $\nabla_{z^t} f$ to be the Euclidean gradient of $f$ w.r.t. $z^t$. To do RSGD in the $H$-tiling model, firstly transform the Euclidean gradient to Riemannian gradient using the pull-back metric:

$$\text{grad}_{z^t} f = z_n^t \nabla_{z^t} f$$

take the learning rate $\eta$ into consideration, denote $v = -\eta \cdot \text{grad}_{z^t} f$, firstly compute its norm as $s = \sqrt{v^T v}$, then the RSGD algorithm in the $H$-tiling model updates $z^t$ as follows:

$$
\begin{cases}
z_i^{t+1} = z_i^t + \dfrac{z_n^t}{\frac{s}{\tanh s} - v_n} \cdot v_i \\
z_n^{t+1} = \dfrac{z_n^t}{\cosh s - \frac{\sinh s}{s} v_n}
\end{cases}
$$

## B  Experiment Details

**Compression** Compression experiments were implemented in Julia. We compress $L$-tiling model using storage methods mentioned in Section 7, round Poincaré ball model towards zero since it is bounded in the Euclidean unit ball, round Lorentz model to the nearest to compress baseline models.

**Learning** We implemented learning experiments in PyTorch using float64. Notably, for tiling-based models, we also use float64 to store the integers, in order to avoid potential numerical imprecision when the integers overflow and are out of the expressible range of float64, we developed a secure method to express a integer matrix $U$ and do accurate integer arithmetic using two float64 type matrices. Specifically, we express it as $U = U_1 + U_2$, where $2^t |U_1|, |U_2| < 2^t$, and $U_1, U_2$ are float64 type. Alternatively, we can similarly use $n$ float64 type matrices to express the integer matrix and pick suitable $t$ to prevent overflow. In our experiments, we found that two float matrices and $t = 20$ are sufficient to prevent overflow and get exact computation of integer arithmetic using float64.

We initialize embeddings randomly from the uniform distribution $U(-0.0001, 0.0001)$, except from embeddings in the $H$-tiling model, whose last elements $z_n$ were initialized from the uniform distribution $U(1, 1.0001)$ in order to make the division to $z_n$ stable. Matrices $g \in G$ in $L$-tiling model were initialized to be identity matrices, integer vectors and the exponential integer in $H$-tiling model were initialized to be zeros. Then we project those embedding to the manifold accordingly before training.

Second, similar as [26], to get a good initial angular layout which is helpful to find good embeddings, we train during an initial "burn-in" phase (20 epochs) with a reduced learning rate $\eta/100$. We train the embedding using multi-threads $N$ to speed up convergence. Those hyperparameters together with batch size $b$ for different datasets were summarized in Table 4.

| Hyperparameters | $\eta$ | $b$ | $N$ |
|---|---|---|---|
| Gr-QC | 0.3 | 10 | 2 |
| WordNet Mammals | 0.3 | 10 | 2 |
| WordNet Verbs | 0.5 | 10 | 5 |
| WordNet Nouns | 0.5 | 50 | 5 |

Table 4: Hyperparameters

## C  More Experiments

Embedding in hyperbolic space reaches better performance compared to Euclidean space, as a simple experiment, consider embedding of a simple tree in Figure 4, where the lengths of edges are in different scale. When embedded into Euclidean space, the global distortion is 0.1395, the worst-case distortion is 2.15. However, when embedded into Poincaré ball model of hyperbolic space, the

Figure 4: A simple tree.

Figure 5: hyperbolic error for bio-yeast dataset.

global distortion is just 0.0007 and the worst-case distortion is 1.025, which is far better than that in Euclidean space.

**Compression Experiments** For compression of Bio-yeast dataset mentioned in Section 7, we plot the scatter of the relationship between log(MSHE) and bits to store per node in Figure 5, under the same MSHE, $L$-tiling model store per node with approximately $3/4$ less bits compared to that of Poincaré model.

We also consider compressing embeddings that were trained using optimization algorithms like SGD and RSGD, to learn 2-dimensional Poincaré disk embedding with great performance, but this method fails to learn a great embedding for larger dataset like WordNet Nouns using 2 dimension model, so we use this method to derive 2-dimensional embeddings for Mammals (Tree-like) and Gr-QC (Dense) dataset. We show the MSHE of different compressions in Table 5, and it leads to the same conclusion that those compression will not hurt the performance of the embedding such as MAP and MR while largely shrinking the size of embeddings.

Table 5: Compression Performance

| Models | Mammals | | | grqc | | |
|---|---|---|---|---|---|---|
| | MSHE | MAP | MR | MSHE | MAP | MR |
| Poincaré(128b) | 0 | 0.7936 | 2.36 | 0 | 0.5382 | 73.88 |
| Poincaré(64b) | 1.59e-2 | 0.7935 | 2.36 | 1.65e-1 | 0.5382 | 73.88 |
| Lorentz(128b) | 1.51e-11 | 0.7935 | 2.36 | 1.39e-10 | 0.5382 | 73.88 |
| Lorentz(64b) | 9.44e-3 | 0.7935 | 2.36 | 1.77e-1 | 0.5382 | 73.88 |
| $L$-tiling-f32 | 5.17e-08 | 0.7935 | 2.36 | 5.29e-8 | 0.5382 | 73.88 |
| $L$-tiling-f16 | 4.16e-04 | 0.7935 | 2.36 | 4.19e-4 | 0.5382 | 73.88 |

**Learning Experiments** We also include the previous results where all models were trained using float32 in PyTorch as shown in Table 6.

# D   Mathematical Background and Proofs

**Definition 1.** *[Representation error] We are concerned with representing points in hyperbolic space $\mathbb{H}^n$ using floating-points* fl. *Define the* representation error *of a particular point $x \in \mathbb{H}^n$ as $\delta_{fl}(x) = d_{\mathbb{H}^n}(x, fl(x))$, and the* worst case representation error *of floating-points representation as a function of the distance-to-origin $d$, which is the maximum representation error of any point with a distance-to-origin at most $d$,*

$$\delta_{fl}^d = \max_{x \in \mathbb{H}^n, \, d_{\mathbb{H}^n}(x,O) \leq d} \delta_{fl}(x).$$

**Definition 3** ([9, 12, 33, 28]). *A tiling of the plane is a collection of sets ("tiles") whose union is the entire plane, but the interiors of different tiles are disjoint. A uniform tiling is an edge-to-edge filling*

Table 6: Previous Learning Experiments

| DIMENSION | MODELS | WORDNET NOUNS | | WORDNET VERBS | | GR-QC | |
|---|---|---|---|---|---|---|---|
| | | MAP | MR | MAP | MR | MAP | MR |
| 2 | POINCARÉ | 0.092 | 95.01 | 0.478 | 6.24 | 0.566 | 69.11 |
| | LORENTZ | 0.371 | 19.07 | 0.701 | 2.35 | 0.556 | **63.62** |
| | $L$-TILING-RSGD | **0.390** | **17.52** | 0.721 | 2.36 | 0.564 | 71.36 |
| | $L$-TILING-SGD | 0.341 | 21.53 | 0.726 | **2.10** | **0.582** | 65.19 |
| | $H$-TILING | 0.385 | 17.70 | **0.741** | 2.28 | 0.568 | 65.84 |
| 4 | 2*$L$-TILING-RSGD | / | / | 0.858 | 1.37 | 0.717 | 12.32 |
| 5 | POINCARÉ | 0.850 | 4.76 | **0.953** | 1.23 | 0.712 | 34.77 |
| | LORENTZ | 0.851 | 3.78 | 0.935 | 1.19 | 0.712 | 33.95 |
| | $H$-TILING | **0.869** | **3.62** | **0.953** | **1.17** | **0.716** | **32.57** |
| 6 | 3*$L$-TILING-RSGD | / | / | 0.935 | 1.13 | **0.852** | **4.45** |
| 10 | POINCARÉ | 0.875 | 3.88 | **0.954** | 1.23 | **0.730** | 29.86 |
| | LORENTZ | 0.872 | 3.45 | 0.951 | **1.14** | 0.724 | 29.50 |
| | $H$-TILING | **0.894** | **3.25** | **0.954** | 1.15 | 0.726 | **29.45** |

*of the hyperbolic plane, which has regular congruent polygons as faces and is vertex-transitive (there is an isometry mapping any vertex onto any other).*

**Definition 4** ([21, 2, 37]). *A Fuchsian group $G$ is a discrete subgroup of the $2\times2$ projective special linear group over $\mathbb{R}$, $PSL(2, \mathbb{R})$.*

**Definition 5** ([37]). *The Dirichlet domain for $G$ centered at $z_0 \in \mathbb{H}^2$ is*

$$\Box(G; z_0) = \{z \in \mathbb{H}^2 : d(z, z_0) \le d(gz, z_0), \forall g \in G\}$$

**Definition 6** ([37, 36]). *A fundamental domain for $G$ is a close set $F \subset \mathbb{H}^2$ such that*

- *$\{gx | \forall g \in G, x \in F\} = \mathbb{H}^2$*

- *$\{gx | \forall x \in F^o\} \cap F^o = \emptyset, \forall g \in G/\{1\}$, where $^o$ denotes the interior.*

**Theorem 1.** *The worst-case representation error (Definition 1) in the Lorentz model using floating-point arithmetic (with machine epsilon $\epsilon_m$) is $\delta_l^d = \text{arcosh}(1 + \epsilon_m(2\cosh^2(d) - 1))$, where $d$ is the hyperbolic distance to origin. This becomes $\delta_l^d = 2d + \log(\epsilon_m) + o(\epsilon_m^{-1}\exp(-2d))$ if $d = O(-\log\epsilon_m)$.*

**Theorem 4.** *The representation error (Definition 1) in L-tiling model is bounded as $\delta_{lt}^d \le \sqrt{5\epsilon_m} + 15\epsilon_m/4 + o(\epsilon_m)$, where $\epsilon_m$ is the machine error.*

*Proof of Theorem 1 and Theorem 4.* For a real point $(g, \boldsymbol{x})$ in $L$-tiling model, where $g \in G$, $\boldsymbol{x} \in F$, we represent it as $(g, \text{fl}(\boldsymbol{x}))$, then we get the representation error as follows:

$$\delta_{lt}^d = d_{lt}((g, \boldsymbol{x}), (g, \text{fl}(\boldsymbol{x}))) = \text{arcosh}(-\boldsymbol{x}^T g^T g_{lt} g \text{fl}(\boldsymbol{x})) = \text{arcosh}(-\boldsymbol{x}^T g_{lt} \text{fl}(\boldsymbol{x}))$$

Note that $|x_i - \text{fl}(x_i)| \le \epsilon_m x_i$, so we have

$$\delta_{lt}^d = \text{arcosh}\left(-(x_1, x_2, x_3)\, g_{lt} \begin{pmatrix} (1+\epsilon_1)x_1 \\ (1+\epsilon_2)x_2 \\ (1+\epsilon_3)x_3 \end{pmatrix}\right)$$
$$= \text{arcosh}((1+\epsilon_1)x_1^2 - (1+\epsilon_2)x_2^2 - (1+\epsilon_3)x_3^2)$$
$$= \text{arcosh}(1 + \epsilon_1 x_1^2 - \epsilon_2 x_2^2 - \epsilon_3 x_3^2)$$

since $x_1^2 + x_2^2 + x_3^2 \le B_f$, then we derive that $\delta_{lt}^d \le \text{arcosh}(1 + \epsilon_m B_f) = \sqrt{\epsilon_m B_f} + 3\epsilon_m B_f/4 + o(\epsilon_m)$, simple calculation will lead to $B_f = 5$, then $\delta_{lt}^d \le \sqrt{5\epsilon_m} + 15\epsilon_m/4 + o(\epsilon_m)$. If we consider the representation error in Lorentz model, the only difference is that $x$ is not bounded in

the fundamental domain any more. Then we can get that $\delta_l^d = \text{arcosh}(1 + \epsilon_m\|x\|^2)$, noticed that $\cosh d = -\boldsymbol{x}^T g_{lt}\boldsymbol{O} = x_1$, where $d$ is the distance to origin, then

$$\delta_l^d = \text{arcosh}(1+\epsilon_m(x_1^2+x_2^2+x_3^2)) = \text{arcosh}(1+\epsilon_m(2x_1^2-1)) = \text{arcosh}(1+\epsilon_m(2\cosh^2(d)-1)),$$

which becomes $\delta_l^d = 2d+\log(\epsilon_m)+o(\epsilon_m^{-1}\exp(-2d))$ if $d = O(-\log\epsilon_m)$, this error also generalize similarly to high dimensional Lorentz model. $\qquad\square$

**Theorem 2.** $F = \{(x_1, x_2, x_3) \in \mathcal{L}^2 | \max(2x_2^2 - x_3^2, 2x_3^2 - x_2^2) < 1\}$ *is a fundamental domain of* $G$. *Any point in* $\mathcal{L}^2$ *can be mapped by* $G$ *to one unique point in* $F$ *or to a point on its boundary.*

*Proof.* To begin with, we prove that $F$ is the Dirichlet domain for $G$ centered at $O \in \mathbb{H}^2$ denoted as $\square(G)$. Firstly, we show that $F \subset \square(G)$, that is, for any $z \in F$, we have $d(z, O) \le d(Uz, O)$ for all $U \in G$. It suffices to show

$$z_1 \le z_1 u_{11} + z_2 u_{12} + z_3 u_{13},$$

where $z_1^2 = 1 + z_2^2 + z_3^2$. Also note that $U \in G$, then $U^T g_l U = g_l$, from which we can derive that $u_{11}^2 = 1 + u_{12}^2 + u_{13}^2$. Further, from the construction of $G$, we can write $u_{11} = t_{11}, u_{12} = \sqrt{3}t_{12}, u_{13} = \sqrt{3}t_{13}$, where $t_{1i}$ is an integer, then $t_{11}^2 = 1 + 3t_{12}^2 + 3t_{13}^2$. Consider following:

$$z_1 \le z_1 u_{11} + z_2 u_{12} + z_3 u_{13}$$

$$\Longleftrightarrow z_1 \le z_1 t_{11} + \sqrt{3}z_2 t_{12} + \sqrt{3}z_3 t_{13}$$

$$\Longleftrightarrow -\sqrt{3}(z_2 t_{12} + z_3 t_{13}) \le z_1(t_{11} - 1)$$

$$\Longleftarrow 3(z_2 t_{12} + z_3 t_{13})^2 \le z_1^2(t_{11}-1)^2 \qquad \triangleright z_1, t_{11} \ge 1$$

$$\Longleftrightarrow 3(z_2^2 t_{12}^2 + z_3^2 t_{13}^2 + 2t_{12}t_{13}z_2z_3) \le (1+z_2^2+z_3^2)(1+3t_{12}^2+3t_{13}^2-2t_{11}+1)$$

$$\Longleftrightarrow 6t_{12}t_{13}z_2z_3 \le (3t_{12}^2+3t_{13}^2-2t_{11}+2)+z_2^2(3t_{13}^2-2t_{11}+2)+z_3^2(3t_{12}^2-2t_{11}+2)$$

$$\Longleftarrow 2t_{11}z_1^2 \le (3t_{12}^2+3t_{13}^2)+2z_1^2 \qquad \triangleright z_2^2 t_{13}^2 + z_3^2 t_{12}^2 \ge 2z_2z_3t_{13}t_{12}$$

$$\Longleftrightarrow 2t_{11}z_1^2 \le (t_{11}^2-1)+2z_1^2$$

$$\Longleftrightarrow 2z_1^2(t_{11}-1) \le t_{11}^2-1$$

$$\Longleftrightarrow 2z_1^2 \le t_{11}+1 \qquad \triangleright t_{11}-1 \ge 0$$

$$\Longleftarrow 5 \le t_{11} \qquad \triangleright z_1 \le \sqrt{3} \Longleftarrow z \in F$$

Hence, if $t_{11} \ge 5$, then the inequality is proved. If $t_{11} < 5$, since $t_{11}$ is an integer, then $t_{11} = 1, 2, 3, 4$.

- If $t_{11} = 1$, then $U$ is identity matrix, the inequality is satisfied.

- If $t_{11} = 2$, then the integer solutions to $t_{11}^2 = 1 + 3t_{12}^2 + 3t_{13}^2$ is $\{g_a, g_b, g_a^{-1}, g_b^{-1}\}$, the inequality is satisfied by simply checking one by one.

- If $t_{11} = 3$, then there is no integer solutions to $t_{11}^2 = 1 + 3t_{12}^2 + 3t_{13}^2$.

- If $t_{11} = 4$, then the solutions to $t_{11}^2 = 1 + 3t_{12}^2 + 3t_{13}^2$ is $(t_{11}, t_{13}, t_{13}) = (4, 2, 1), (4, 1, 2)$. Manually check will find that the inequality is satisfied in both cases.

Then $F \subset \square(G)$, also note that with Algorithm 1, for any $z \in \square(G)$, it will be mapped by some $V \in G$ such that $Vz \in F$ and $d(Vz, O) \le d(z, O)$. From the definition of $\square(G)$, $d(Vz, O) \ge d(z, O)$, then $(Vz)^T g_l O = d(Vz, O) = d(z, O) = z^T g_l O$, which leads to $V_{11} = 1$, then $V = I$ considering $V^T g_l V = g_l$, thus, $z = Vz \in F$ and $\square(G) \subset F$, which shows that $F = \square(G)$.

For the second part of the proof, we show that $F$ is a fundamental domain for $G$. According to Theorem 37.1.10 in [37], it suffices to show that $Stab_G(O) = \{1\}$. Consider $TO_H = O_H$, where $T \in G$. From $(T - I)O_H = 0$, we can get that

$$T = \begin{bmatrix} 1 & 0 \\ 0 & B \end{bmatrix}.$$

where $B^T = B$. Also note $T \in G$, then $B^2 = I$, these two conditions lead to that $B = I$. Hence, $Stab_G(O) = \{1\}$, then $F$ is a fundamental domain for $G$. Since fundamental domain $F$ only contains one element in the orbit, then for any point in the space, it can only be mapped to one unique point in $F$. $\qquad\square$

**Theorem 3.** *For any point in the Lorentz model, Algorithm 1 converges and stops within $1 + 7d$ steps, where $d = d(\boldsymbol{x}, \boldsymbol{O})$ denotes the distance from $\boldsymbol{x}$ to the origin.*

*Proof.* We just consider $x_2 \leq -|x_3|$ case in the Algorithm, other cases can be proved in the same way, then we have

$$\frac{\cosh d(L \cdot g_a \cdot L^{-1} \cdot \boldsymbol{x}, O)}{\cosh d(\boldsymbol{x}, \boldsymbol{O})} = 2 + \frac{\sqrt{3}x_2}{\sqrt{1 + x_2^2 + x_3^2}} < 1$$

then the distance to the origin in the space is monotonically decreasing as Algorithm 1 goes, note that this distance is bounded by 0, then it will converge.

To see the steps required for the algorithm to finish, we may assume that $\max\{|x_2|, |x_3|\} \geq C_0 > 1$, due to the symmetry of this algorithm, also consider the case $x_2 \leq -|x_3|$, then $|x_2| \geq C_0$, we have

$$\frac{\cosh d(L \cdot g_a \cdot L^{-1} \cdot \boldsymbol{x}, \boldsymbol{O})}{\cosh d(\boldsymbol{x}, \boldsymbol{O})} \leq 2 - \sqrt{\frac{3C_0}{2C_0 + 1}} \leq 1$$

Hence,

$$d(\boldsymbol{x}, \boldsymbol{O}) - d(L \cdot g_a \cdot L^{-1} \cdot \boldsymbol{x}, \boldsymbol{O})$$

$$\geq \operatorname{arcosh}(\sqrt{1 + x_2^2 + x_3^2}) - \operatorname{arcosh}\left( (2 - \sqrt{\frac{3C_0}{2C_0 + 1}})\sqrt{1 + x_2^2 + x_3^2} \right)$$

$$\geq -\log(2 - \sqrt{\frac{3C_0}{2C_0 + 1}})$$

so $d(\boldsymbol{x}, \boldsymbol{O})$ will decrease monotonically for at most $s_0(C_0)$ steps, where

$$s_0(C_0) = \frac{d(\boldsymbol{x}, \boldsymbol{O})}{-\log(2 - \sqrt{\frac{3C_0}{2C_0+1}})}$$

Consider $\max\{|x_2|, |x_3|\}$ at the boundary between $F$ and its neighborhood tiles:

$$\min_{x \in L\{g_a, g_b, g_a^{-1}, g_b^{-1}\}L^{-1}F} \max\{|x_2|, |x_3|\} = \frac{5}{2}\sqrt{2}$$

Hence, we can choose $C_0 = \frac{5}{2}\sqrt{2}$, then $d(\boldsymbol{x}, \boldsymbol{O})$ will decrease monotonically until $\max\{|x_2|, |x_3|\} < C_0$ within $s_0(5\sqrt{2}/2)$ steps, which means $x$ lies either in $F$ or its 4 neighborhood tiles, so totally it will cost at most $s_0(5\sqrt{2}/2) + 1 \leq 1 + 7d$ steps. $\qquad\square$

**Lemma 1.** *If the integer matrix $U$ is given, then corresponding VBW encoding can be derived using Algorithm 1. Further, if only $U_{21}, U_{31}$ are given, then $U$ can be reconstructed using Algorithm 1.*

*Proof.* If $U$ is given, consider the point $(U, O)$ in the $L$-tiling model, which is in correspondence to $x = LUL^{-1}O$ in the Lorentz model, then we can map $x$ to $(U', u')$ with Algorithm 1, where we choose a generator at each step to get a generator order string, with which we can reconstruct $U'$. Since each point in the Lorentz model will be mapped to a unique point in $F$ as Theorem 2 states, also $x$ can be mapped to $(U, O)$ and $(U', u')$, then $u' = O$. The question is whether $U = U'$, consider $LUL^{-1}O = LU'L^{-1}O$, which leads to $(LU'^{-1}UL^{-1} - I)O = 0$, since $Stab_G(O) = \{1\}$ as the second part in the proof of Theorem 2 proved, then $LU'^{-1}UL^{-1} = I$ to get $U = U'$. Hence, given $U$, we can get its generator order string, which can be then used to get the VBW encoding accordingly. Further, note that $U^T M_3 U = M_3$, then we have

$$U_{11}^2 = 1 + \frac{U_{21}^2 + U_{31}^2}{3}.$$

Therefore, we can compute $U_{11}$ if only $U_{21}, U_{31}$ were given to get the first column of $U$. Since $x = LUL^{-1}O$ and $O = (1, 0, 0)$, then the first column of $U$ suffices to get $x$, then we can reconstruct $U$ out using Algorithm 1. $\qquad\square$

**Lemma 2.** *$Q_{11}$ has the largest absolute value in $Q = U^T M_3 V$.*

*Proof.* Note that $Q = U^T M_3 V = M_3 U^{-1} V = M_3 T$, where $T$ is an integer matrix generated by $g_a$ and $g_b$, so we have $T^T M_3 T = M_3$, using this relation, we can get following equations:

$$t_{11}^2 = 1 + \frac{t_{21}^2 + t_{31}^2}{3}$$
$$t_{11}^2 = 1 + 3(t_{12}^2 + t_{13}^2)$$
$$3t_{12}^2 = (t_{22}^2 + t_{32}^2) - 1$$
$$3t_{13}^2 = (t_{23}^2 + t_{33}^2) - 1$$

since $Q_{11} = 3t_{11}$, from first formula, we get that $Q_{11}^2 = 9t_{11}^2 \geq 3t_{11}^2 > t_{21}^2 + t_{31}^2 = Q_{21}^2 + Q_{31}^2$, so $Q_{11}$ has the largest absolute value in the first column of $Q$. From the second formula, we get that $Q_{11}^2 \geq t_{11}^2 > t_{12}^2 + t_{13}^2$, so $Q_{11}$ has the largest absolute value in the first row of $Q$. Then combine formulas 2,3,4 and we get that

$$1 + t_{11}^2 = t_{22}^2 + t_{32}^2 + t_{23}^2 + t_{33}^2$$

From first formula, we know that $t_{11} \geq 1$, then we have $Q_{11}^2 \geq 2t_{11}^2 \geq 1 + t_{11}^2 = t_{22}^2 + t_{32}^2 + t_{23}^2 + t_{33}^2$ Combine above results, clearly that $Q_{11}$ has the largest absolute value in $Q$, which finishes our proof $\qquad\square$

**Theorem 5.** *The representation error (Definition 1) in H-tiling model is bounded as $\delta_{ht}^d = \sqrt{(n+3)\epsilon_m}/2 + (n+3)\epsilon_m/4 + o(\epsilon_m)$, where $\epsilon_m$ is the machine error.*

*Proof.* For a real point $(j, \boldsymbol{k}, \boldsymbol{x}) \in \mathbb{Z} \times (\mathbb{Z}^{n-1} \times \{0\}) \times S$ in $H$-tiling model, we represent it as $(j, \boldsymbol{k}, \mathrm{fl}(\boldsymbol{x}))$, then we get the representation error as follows:

$$\delta_{ht}^d = d_{ht}((j, \boldsymbol{k}, \boldsymbol{x}), (j, \boldsymbol{k}, \mathrm{fl}(\boldsymbol{x}))) = \mathrm{arcosh}(1 + \frac{||\boldsymbol{x} - \mathrm{fl}(\boldsymbol{x}))||^2}{2x_n \mathrm{fl}(x_n)})$$

Note that $|x_i - \mathrm{fl}(x_i)| \leq \epsilon_m x_i$, so we have

$$\delta_{ht}^d = \mathrm{arcosh}(1 + \frac{\sum\limits_{i=1}^{n} \epsilon_i x_i^2}{2(1+\epsilon_n)x_n^2})$$

since $0 \leq x_1, \cdots, x_{n-1} < 1 \leq x_n < 2$, then we derive that $\delta_{ht}^d \leq \mathrm{arcosh}(1 + \frac{(n+3)\epsilon_m}{2(1-\epsilon_m)}) = \sqrt{(n+3)\epsilon_m}/2 + (n+3)\epsilon_m/4 + o(\epsilon_m)$. $\qquad\square$

*Proof of RSGD algorithm.* Here we show the equivalence between the RSGD algorithm of $L$-tiling model described in appendix A and that in Lorentz model. To begin with, consider the RSGD algorithm in Lorentz model. Let $x_t, y_t \in \mathcal{H}^2$, then we have $d(x_t, y_t) = \mathrm{arcosh}(-x_t^T g_l y_t)$, the Euclidean gradient of $x_t$ can be computed as

$$\nabla_{x_t} d(x_t, y_t) = \frac{-g_l y_t}{\sqrt{(x_t^T g_l y_t)^2 - 1}},$$

to get the Riemannian gradient in the model, we make use of the pull-back metric as follows,

$$h_t = g_{l,x_t}^{-1} \nabla_{x_t} d(x_t, y_t) = \frac{-y_t}{\sqrt{(x_t^T g_l y_t)^2 - 1}},$$

further we project this Riemannian gradient into the tangent space at $x_t$,

$$\mathrm{grad}_{x_t} d = h_t + \langle x_t, h_t \rangle_L x_t = -\frac{y_t + (x_t^T g_l y_t)x_t}{\sqrt{(x_t^T g_l y_t)^2 - 1}},$$

then we make use of the exponential map in Lorentz model to update,

$$x_{t+1} = \exp_{x_t}(v) = \cosh(||v||_L)x_t + \sinh(||v||_L)\frac{v}{||v||_L},$$

where
$$v = -\eta \cdot \mathrm{grad}_{x_t} d = \eta \frac{y_t + (x_t^T g_l y_t)x_t}{\sqrt{(x_t^T g_l y_t)^2 - 1}}.$$

Now consider the norm of $v$ under hyperbolic metric,

$$\|v\|_L^2 = v^T g_l v$$
$$= \frac{\eta^2}{(x_t^T g_l y_t)^2 - 1}(y_t^T g_l y_t + (x_t^T g_l y_t)^2 + (x_t^T g_l y_t)^2 + (x_t^T g_l y_t)^2 x_t^T g_l x_t)$$
$$= \frac{\eta^2}{(x_t^T g_l y_t)^2 - 1}(-1 + (x_t^T g_l y_t)^2 + (x_t^T g_l y_t)^2 - (x_t^T g_l y_t)^2)$$
$$= \eta^2$$

hence we derived the RSGD algorithm in the Lorentz model as

$$x_{t+1} = \exp_{x_t}(v) = \cosh(\eta)x_t + \sinh(\eta)\frac{y_t + (x_t^T g_l y_t)x_t}{\sqrt{(x_t^T g_l y_t)^2 - 1}} \tag{2}$$

For the second part, we turn to $L$-tiling model, let $x_t = LUL^{-1}u_t, y_t = LVL^{-1}v_t$, the distance is

$$d(x_t, y_t) = \mathrm{arcosh}(u_t^T L^{-T} Q L^{-1} v_t),$$

then the Euclidean gradient of $u_t$ can be computed as

$$\nabla_{u_t} d(x, y) = \frac{L^{-T} Q L^{-1} v_t}{\sqrt{(u_t^T L^{-T} Q L^{-1} v_t)^2 - 1}}.$$

In the same way, make use of the pull-back by the metric matrix, we derived the Riemannian gradient

$$h_t = g_{lt,u_t}^{-1} \nabla_{u_t} d(x_t, y_t) = \frac{g_{lt} L^{-T} Q L^{-1} v_t}{\sqrt{(u_t^T L^{-T} Q L^{-1} v_t)^2 - 1}},$$

also project it into the tangent space at $u_t$,

$$\mathrm{grad}_{u_t} d = h_t + \langle u_t, h_t \rangle_L u_t = \frac{g_{lt} L^{-T} Q L^{-1} v_t + (v_t^T L^{-T} Q^T L^{-1} u_t)u_t}{\sqrt{(u_t^T L^{-T} Q L^{-1} v_t)^2 - 1}}.$$

then the update rule in the tiling-based model is

$$u_{t+1} = \exp_{u_t}(v),$$

where
$$v = -\eta \cdot \mathrm{grad}_{u_t} d = -\eta \frac{g_{lt} L^{-T} Q L^{-1} v_t + (v_t^T L^{-T} Q^T L^{-1} u_t)u_t}{\sqrt{(u_t^T L^{-T} Q L^{-1} v_t)^2 - 1}},$$

then consider the norm of $v$ under hyperbolic metric,

$$\|v\|_L^2 = v^T g_{lt} v$$
$$= \frac{\eta^2}{(u_t^T L^{-T} Q L^{-1} v_t)^2 - 1}[g_{lt} L^{-T} Q L^{-1} v_t + (v_t^T L^{-T} Q^T L^{-1} u_t)u_t]^T \cdot$$
$$g_{lt}[g_{lt} L^{-T} Q L^{-1} v_t + (v_t^T L^{-T} Q^T L^{-1} u_t)u_t]$$
$$= \frac{\eta^2}{(u_t^T L^{-T} Q L^{-1} v_t)^2 - 1}[v_t^T L^{-T} Q^T L^{-1} g_{lt} L^{-T} Q L^{-1} v_t$$
$$+ 2(v_t^T L^{-T} Q^T L^{-1} u_t)^2 + (v_t^T L^{-T} Q^T L^{-1} u_t)u_t^T g_{lt} u_t]$$
$$= \frac{\eta^2}{(u_t^T L^{-T} Q L^{-1} v_t)^2 - 1}[v_t^T L^{-T} Q^T L^{-1} g_{lt} L^{-T} Q L^{-1} v_t$$
$$+ 2(v_t^T L^{-T} Q^T L^{-1} u_t)^2 - (v_t^T L^{-T} Q^T L^{-1} u_t)]$$
$$= \frac{\eta^2}{(u_t^T L^{-T} Q L^{-1} v_t)^2 - 1}[(L^{-T} Q L^{-1} v_t)^T g_{lt}(L^{-T} Q L^{-1} v_t) + (v_t^T L^{-T} Q^T L^{-1} u_t)^2]$$

Since $U \in G_0$, then it follows

$$Q = U^T M_3 V = M_3 U^{-1} V = M_3 W, \quad W \in G.$$

So we get

$$L^{-T} Q L^{-1} v_t = L^{-T} M_3 W L^{-1} v_t,$$

hence

$$
\begin{aligned}
(L^{-T} Q L^{-1} v_t)^T g_{lt} (L^{-T} Q L^{-1} v_t) &= v_t^T L^{-T} W^T M_3^T L^{-1} g_{lt} L^{-T} M_3 W L^{-1} v_t \\
&= - v_t^T L^{-T} W^T M_3 W L^{-1} v_t \\
&= - v_t^T L^{-T} M_3 L^{-1} v_t \\
&= v_t^T g_{lt} v_t \\
&= -1
\end{aligned}
$$

so $\|v\|_L^2 = \eta^2$, then the RSGD algorithm in $L$-tiling model is

$$u_{t+1} = \exp_{u_t}(v) = \cosh(\eta) u_t - \sinh(\eta) \frac{g_{lt} L^{-T} Q L^{-1} v_t + (v_t^T L^{-T} Q^T L^{-1} u_t) u_t}{\sqrt{(u_t^T L^{-T} Q L^{-1} v_t)^2 - 1}} \qquad (3)$$

For the third part, again consider the RSGD algorithm in Lorentz model, from Equation 2, we have that

$$LUL^{-1} u_{t+1} = \cosh(\eta) LUL^{-1} u_t + \sinh(\eta) \frac{LVL^{-1} v_t - (v_t^T L^{-T} Q^T L^{-1} u_t) LUL^{-1} u_t}{\sqrt{(v_t^T L^{-T} Q^T L^{-1} u_t)^2 - 1}},$$

so RSGD algorithm in Lorentz model is equivalent to:

$$u_{t+1} = \cosh(\eta) u_t + \sinh(\eta) \frac{LU^{-1} V L^{-1} v_t - (v_t^T L^{-T} Q^T L^{-1} u_t) u_t}{\sqrt{(v_t^T L^{-T} Q^T L^{-1} u_t)^2 - 1}}, \qquad (4)$$

note that $U^T M_3 U = M_3$, then $U^{-1} = M_3^{-1} U^T M_3$, with simple computation, we get that

$$LU^{-1} V L^{-1} = L M_3^{-1} U^T M_3 V L^{-1} = L M_3^{-1} Q L^{-1} = -g L^{-T} Q L^{-1},$$

hence, RSGD algorithm in $L$-tiling model (Equation 3) becomes the same as RSGD algorithm in Lorentz model (Equation 4), which finishes our proof. $\qquad\square$

**Error for Computing in $L$-tiling Model** We approximate $(U, u), (V, v)$ with $(U, \mathrm{fl}(u))$ and $(V, \mathrm{fl}(v))$, here we provide the error of computing in $L$-tiling model together with that in Lorentz model.

**Theorem 6.** *The worst case error of computing distance in Lorentz model using float is*

$$|d_l^{\mathit{fl}}(\mathit{fl}(x), \mathit{fl}(y)) - d_l(x, y)| = \frac{2 \cosh(d(x, O)) \cosh(d(y, O))}{\sinh(d(x, y))} \epsilon_m + \epsilon_m d(x, y) + o(\epsilon_m),$$

*the error of computing distance in $L$-tiling model using float is*

$$|d_{lt}^{\mathit{fl}}((U, \mathit{fl}(u)), (V, \mathit{fl}(v))) - d_{lt}((U, u), (V, v))| = \begin{cases} d\epsilon_m + A_1(C_0)\epsilon_m + o(C_0^{-2} + \epsilon_m), d \geq C_0 \\ d\epsilon_m + [A_2(C_0) + \dfrac{A_3(C_0)}{\tanh(d)}]\epsilon_m + o(\epsilon_m), d < C_0 \end{cases}$$

*where $d$ is the real distance between two points, $A_i(C_0)$ are constants only depends on $C_0$, $d^{\mathit{fl}}$ means that inside computation like multiplication are performed with machine error $\epsilon_m$.*

**Remark**: The worst case error of distance computation in Lorentz model using float is dominated by $d(x, O), d(y, O)$, this will cause the "NaN" problem when two points are far away from the origin. However, in $L$-tiling model, the error only depends on $d$, i.e., how far two points are to each other, also $\tanh$ term is bounded, which controls the distance error and solves the "NaN" problem.

*Proof.* We consider the Lorentz model at first, let $z = x^T M y$, $\hat{z} = \text{fl}(x)^T M \text{fl}(y)$, then

$$|\hat{z} - z| \leq |(1 + \epsilon_m)^6 z - z| = 6\epsilon_m |x|^T |y| + o(\epsilon_m) \leq 2\epsilon_m x_0 y_0 + o(\epsilon_m)$$
$$= 2\epsilon_m \cosh(d_x) \cosh(d_y) + o(\epsilon_m) = \delta_z$$

thus,

$$|d_l^{\text{fl}}(\text{fl}(x), \text{fl}(y)) - d_l(x, y)| = |(1 + \epsilon) \operatorname{arcosh}(x^T M y + \delta_z) - \operatorname{arcosh}(x^T M y)|$$
$$= \frac{\delta_z}{\sqrt{(x^T M y)^2 - 1}} + \epsilon d_l + o(\delta_z) = \frac{2 \cosh(d_x) \cosh(d_y)}{\sinh(d_l)} \epsilon_m + \epsilon_m d_l + o(\epsilon_m)$$

As for the distance error in $L$-tiling model, here in the same way denote $z = h(u)^T L^{-T} \hat{Q} L^{-1} h(v)$, since $h(u), h(v)$ are in the fundamental domain, which is bounded by $B_f$, so $\|h(u)\|, \|h(v)\| \leq \sqrt{B_f} = \sqrt{5}$, also $\hat{Q}$ is bounded, then using Cauchy inequality, we have

$$|z| \leq \|h(u)\| (\frac{\sqrt{7}}{3} |h(v)_1| + \sqrt{\frac{7}{3}} |h(v)_2| + \sqrt{\frac{7}{3}} |h(v)_3|) \leq \frac{7}{3} \|h(u)\| \|h(v)\| \leq \frac{7}{3} B_f^2 = 35/3$$

so the distance is

$$d_{lt}((U, u), (V, v)) = \log(Q_{11}) + \log\left(z + \sqrt{z^2 - Q_{11}^{-2}}\right)$$

then $\log(Q_{11}) \leq d$, further note that

$$\operatorname{arcosh}(Q_{11}/3) = d(L^{-T} U L^{-1} O, L^{-T} V L^{-1} O)$$
$$\leq d(L^{-T} U L^{-1} O, L^{-T} U L^{-1} h(u)) + d(L^{-T} U L^{-1} h(u), L^{-T} V L^{-1} h(v))$$
$$\quad + d(L^{-T} V L^{-1} O, L^{-T} V L^{-1} h(v))$$
$$= d(O, h(u)) + d(L^{-T} U L^{-1} h(u), L^{-T} V L^{-1} h(v)) + d(O, h(v))$$
$$\leq d + 2 \operatorname{arcosh}(\sqrt{3}) = d + 2 \log(\sqrt{3} + \sqrt{2})$$

in this way, we get

$$z^2 = z^2 - Q_{11}^{-2} + Q_{11}^{-2} = Q_{11}^{-2}(\cosh^2(d) - 1) + Q_{11}^{-2} \geq Q_{11}^{-2} + \frac{\sinh^2(d)}{9 \cosh^2(d + 2B_f)}$$
$$z = Q_{11}^{-1} \cosh(d) \geq \frac{\cosh(d)}{3 \cosh(d + 2 \log(\sqrt{3} + \sqrt{2}))}$$

Now, we consider the first term of calculating distance $\log(Q_{11})$, in order to avoid overflow, we computed with following formula.

$$\log(Q_{11}) = \frac{\log(3)}{2} + \log(U^{11}) + \log(V_{11}) + \log(1 + \frac{U^{12}}{U^{11}} \frac{V_{12}}{V_{11}} + \frac{U^{13}}{U^{11}} \frac{V_{13}}{V_{11}})$$

then

$$\text{flc}(\log(Q_{11})) = \left(\frac{\log(3)}{2} + \log(U^{11}) + \log(V_{11})\right)(1 + \epsilon)$$

$$+ \text{fl}\left(\log(1 + \frac{U^{12}}{U^{11}}\frac{V_{12}}{V_{11}}(1 + \epsilon)^2 + \frac{U^{13}}{U^{11}}\frac{V_{13}}{V_{11}}(1 + \epsilon)^2)\right)$$

$$= \left(\frac{\log(3)}{2} + \log(U^{11}) + \log(V_{11})\right)(1 + \epsilon)$$

$$+ \left(\log(1 + \frac{U^{12}}{U^{11}}\frac{V_{12}}{V_{11}}(1 + \epsilon)^2 + \frac{U^{13}}{U^{11}}\frac{V_{13}}{V_{11}}(1 + \epsilon)^2)\right)(1 + \epsilon)$$

$$= \left(\frac{\log(3)}{2} + \log(U^{11}) + \log(V_{11})\right)(1 + \epsilon)$$

$$+ (1 + \epsilon)\log(1 + \frac{U^{12}}{U^{11}}\frac{V_{12}}{V_{11}}(1 + 2\epsilon) + \frac{U^{13}}{U^{11}}\frac{V_{13}}{V_{11}}(1 + 2\epsilon) + o(\epsilon))$$

$$= \left(\frac{\log(3)}{2} + \log(U^{11}) + \log(V_{11})\right)(1 + \epsilon) + (1 + \epsilon)\log(1 + \frac{U^{12}}{U^{11}}\frac{V_{12}}{V_{11}} + \frac{U^{13}}{U^{11}}\frac{V_{13}}{V_{11}})$$

$$+ (1 + \epsilon)\frac{2\epsilon(\frac{U^{12}}{U^{11}}\frac{V_{12}}{V_{11}} + \frac{U^{13}}{U^{11}}\frac{V_{13}}{V_{11}}) + o(\epsilon)}{1 + \frac{U^{12}}{U^{11}}\frac{V_{12}}{V_{11}} + \frac{U^{13}}{U^{11}}\frac{V_{13}}{V_{11}}}$$

$$= \log(Q_{11})(1 + \epsilon) + (1 + \epsilon)\left(\frac{2\epsilon(\frac{U^{12}}{U^{11}}\frac{V_{12}}{V_{11}} + \frac{U^{13}}{U^{11}}\frac{V_{13}}{V_{11}}) + o(\epsilon)}{1 + \frac{U^{12}}{U^{11}}\frac{V_{12}}{V_{11}} + \frac{U^{13}}{U^{11}}\frac{V_{13}}{V_{11}}}\right)$$

$$= \log(Q_{11})(1 + \epsilon) + 2\epsilon\frac{U^{12}V_{12} + U^{13}V_{13}}{U^{11}V_{11} + U^{12}V_{12} + U^{13}V_{13}} + o(\epsilon)$$

Here flc means calculating with float arithmetic, hence, the error of computing the first term is

$$\delta_Q = |\text{flc}(\log(Q_{11})) - \log(Q_{11})| \le \log(Q_{11})\epsilon_m + \frac{1}{2}\epsilon_m + o(\epsilon_m)$$

Then we consider the error of computing $z = h(u)^T L^{-T}\hat{Q}L^{-1}h(v)$, given by following formula:

$$|\textbf{flc}(z) - z| \le |(1 + \epsilon)^7 h(u)^T L^{-T}\hat{Q}L^{-1}h(v) - z| \le 7\epsilon_m|h(u)|^T L^{-T}|\hat{Q}|L^{-1}|h(v)| + o(\epsilon_m)$$

$$\le \frac{245}{3}\epsilon_m + o(\epsilon_m) = \delta_z$$

based on this error, we consider the error for the second term of distance

$$\delta_2 = \text{flc}\left(\log(z + \sqrt{z^2 - Q_{11}^{-2}})\right) - \log(z + \sqrt{z^2 - Q_{11}^{-2}})$$

$$= (1 + \epsilon_1)(\log((1 + \epsilon_2)(z + \delta_z + (1 + \epsilon_3)\sqrt{(1 + \epsilon_4)((1 + \epsilon_5)(z + \delta_z)^2 - (1 + \epsilon_6)Q_{11}^{-2})})))$$

$$- \log(z + \sqrt{z^2 - Q_{11}^{-2}})$$

$$= (1 + \epsilon_1)(\log(z + \delta_z + (1 + \epsilon_3)\sqrt{(1 + \epsilon_4)((1 + \epsilon_5)(z + \delta_z)^2 - (1 + \epsilon_6)Q_{11}^{-2})}))$$

$$- \log(z + \sqrt{z^2 - Q_{11}^{-2}}) + \epsilon_2 + o(\epsilon_m)$$

$$= (1 + \epsilon_1)(\log(z + \delta_z + (1 + \epsilon_3)\sqrt{(1 + \epsilon_4)} \cdot \sqrt{(z + \delta_z)^2 - Q_{11}^{-2} + \epsilon_7((z + \delta_z)^2 + Q_{11}^{-2})}))$$

$$- \log(z + \sqrt{z^2 - Q_{11}^{-2}}) + \epsilon_2 + o(\epsilon_m)$$

$$= (1 + \epsilon_1)(\log(z + \frac{245}{3}\epsilon_m + (1 + \frac{2}{3}\epsilon_8)\sqrt{(z + \frac{245}{3}\epsilon_m)^2 - Q_{11}^{-2} + \epsilon_7((z + \delta_z)^2 + Q_{11}^{-2})}))$$

$$- \log(z + \sqrt{z^2 - Q_{11}^{-2}}) + \epsilon_2 + o(\epsilon_m)$$

We divide it into two cases, firstly, consider Taylor expansion here when $Q_{11}z \geq C_0$, where $C_0 \geq 1$ is a large constant, then

$$\log(z + \frac{245}{3}\epsilon_m + (1 + \frac{2}{3}\epsilon_8)\sqrt{(z + \frac{245}{3}\epsilon_m)^2 - Q_{11}^{-2} + \epsilon_7((z + \frac{245}{3}\epsilon_m)^2 + Q_{11}^{-2})})$$

$$= \log(z + (1 + \frac{2}{3}\epsilon_8)\sqrt{z^2 - Q_{11}^{-2} + \epsilon_7(z^2 + Q_{11}^{-2})}) + \frac{245\epsilon_m}{3\sqrt{z^2 - Q_{11}^{-2} + \epsilon_7(z^2 + Q_{11}^{-2})}} + o(\epsilon_m)$$

$$= \log(z + \sqrt{z^2 - Q_{11}^{-2} + \epsilon_7(z^2 + Q_{11}^{-2})}) + \frac{2}{3}\epsilon_8 - \frac{2\epsilon_8 z}{3[z + \sqrt{z^2 - Q_{11}^{-2} + \epsilon_7(z^2 + Q_{11}^{-2})}]}$$

$$+ \frac{245\epsilon_m}{3\sqrt{z^2 - Q_{11}^{-2} + \epsilon_7(z^2 + Q_{11}^{-2})}} + o(\epsilon_m)$$

$$= \log(z + \sqrt{z^2 - Q_{11}^{-2}}) + \frac{(z^2 + Q_{11}^{-2})\epsilon_7}{2\sqrt{z^2 - Q_{11}^{-2}}(z + \sqrt{z^2 - Q_{11}^{-2}})} + \frac{2}{3}\epsilon_8 - \frac{2\epsilon_8 z}{3[z + \sqrt{z^2 - Q_{11}^{-2}}]}$$

$$+ \frac{245\epsilon_m}{3\sqrt{z^2 - Q_{11}^{-2}}} + o(\epsilon_m)$$

$$= \log(z + \sqrt{z^2 - Q_{11}^{-2}}) + \frac{\epsilon_7}{4} + \frac{7\epsilon_7}{16(Q_{11}z)^2} + \frac{2}{3}\epsilon_8 - \frac{\epsilon_8}{3} - \frac{\epsilon_8}{12(Q_{11}z)^2} + \frac{245\epsilon_m}{3z}$$

$$+ O(\epsilon_m Q_{11}^{-2}z^{-3}) + o((Q_{11}z)^{-3}) + o(\epsilon_m)$$

$$= \log(z + \sqrt{z^2 - Q_{11}^{-2}}) + \frac{\epsilon_7}{4} + \frac{7\epsilon_7}{16(Q_{11}z)^2} + \frac{1}{3}\epsilon_8 - \frac{\epsilon_8}{12(Q_{11}z)^2} + \frac{245\epsilon_m}{3z}$$

$$+ O(\epsilon_m Q_{11}^{-2}z^{-3}) + o((Q_{11}z)^{-3}) + o(\epsilon_m)$$

$$= \log(z + \sqrt{z^2 - Q_{11}^{-2}}) + \frac{\epsilon_7}{4} + \frac{7\epsilon_7}{16(Q_{11}z)^2} + \frac{1}{3}\epsilon_8 - \frac{\epsilon_8}{12(Q_{11}z)^2} + \frac{245\epsilon_m}{3z} + o(C_0^{-2}) + o(\epsilon_m)$$

$$= \log(z + \sqrt{z^2 - Q_{11}^{-2}}) + (\frac{7}{12} + \frac{25}{48(Q_{11}z)^2})\epsilon_9 + \frac{245\epsilon_m}{3z} + o(C_0^{-2}) + o(\epsilon_m)$$

where $|\epsilon_i| \leq \epsilon_m$, the machine error, then the error is

$$|(1 + \epsilon_1)(\log(z + \sqrt{z^2 - Q_{11}^{-2}}) + (\frac{7}{12} + \frac{25}{48(Q_{11}z)^2})\epsilon_9 + \frac{245\epsilon_m}{3z} + o(C_0^{-2}) + o(\epsilon_m))$$

$$- \log(z + \sqrt{z^2 - Q_{11}^{-2}}) + \epsilon_2 + o(\epsilon_m)|$$

$$= |(1 + \epsilon_1)\left((\frac{7}{12} + \frac{25}{48(Q_{11}z)^2})\epsilon_9 + \frac{245\epsilon_m}{3z} + o(C_0^{-2})\right) + \epsilon_1 \log(z + \sqrt{z^2 - Q_{11}^{-2}}) + \epsilon_2 + o(\epsilon_m)|$$

$$= |(\frac{7}{12} + \frac{25}{48(Q_{11}z)^2})\epsilon_9 + \frac{245\epsilon_m}{3z} + \epsilon_1 \log(z + \sqrt{z^2 - Q_{11}^{-2}}) + \epsilon_2 + o(C_0^{-2}) + o(\epsilon_m) + \frac{7}{12}\epsilon_1|$$

$$\leq \frac{277}{432}\epsilon_m + 63\epsilon_m C_0 Q_{11}^{-1} + \epsilon_m \log 70/3 + \frac{19}{12}\epsilon_m + o(C_0^{-2}) + o(\epsilon_m)$$

$$\leq (\frac{961}{432} + \frac{245C_0}{3} + \log 70/3)\epsilon_m + o(C_0^{-2}) + o(\epsilon_m) = \delta_2$$

On the other hand, if $Q_{11}z \leq C_0$, then notice that

$$\text{arcosh}(Q_{11}/3) = d(L^{-T}UL^{-1}O, L^{-T}VL^{-1}O)$$

$$\leq d(L^{-T}UL^{-1}O, L^{-T}UL^{-1}h(u)) + d(L^{-T}UL^{-1}h(u), L^{-T}VL^{-1}h(v))$$

$$+ d(L^{-T}VL^{-1}O, L^{-T}VL^{-1}h(v))$$

$$= d(O, h(u)) + d(L^{-T}UL^{-1}h(u), L^{-T}VL^{-1}h(v)) + d(O, h(v))$$

$$\leq \text{arcosh}(Q_{11}z) + 2\log(\sqrt{3} + \sqrt{2})$$

$$\leq \text{arcosh}(C_0) + 2\log(\sqrt{3} + \sqrt{2})$$

so we can get $Q_{11} \le E(C_0)$, where $E(C_0)$ is a constant depending on $C_0$, then we further have

$$\log(z + \frac{245}{3}\epsilon_m + (1 + \frac{2}{3}\epsilon_8) \cdot \sqrt{(z + \frac{245}{3}\epsilon_m)^2 - Q_{11}^{-2} + \epsilon_7((z + \frac{245}{3}\epsilon_m)^2 + Q_{11}^{-2})})$$

$$= \log(Q_{11}z + \frac{245}{3}\epsilon_m Q_{11} - \log(Q_{11})$$

$$+ (1 + \frac{2}{3}\epsilon_8) \cdot \sqrt{(Q_{11}z + \frac{245}{3}\epsilon_m Q_{11})^2 - 1 + \epsilon_7((Q_{11}z + \frac{245}{3}\epsilon_m Q_{11})^2 + 1)})$$

$$= \log(Q_{11}z + (1 + \frac{2}{3}\epsilon_8)\sqrt{(Q_{11}z)^2 - 1 + \epsilon_7((Q_{11}z)^2 + 1)}) - \log(Q_{11}) + o(\epsilon_m)$$

$$+ \frac{245}{3}\epsilon_m Q_{11}[1 + \frac{Q_{11}z}{\sqrt{(Q_{11}z)^2 - 1}}]\frac{1}{Q_{11}z + \sqrt{(Q_{11}z)^2 - 1}}$$

$$= \log(Q_{11}z + \sqrt{(Q_{11}z)^2 - 1 + \epsilon_7((Q_{11}z)^2 + 1)}) - \log(Q_{11}) + o(\epsilon_m)$$

$$+ (\frac{245}{3}\epsilon_m Q_{11} + \frac{2}{3}\epsilon_8)[1 + \frac{Q_{11}z}{\sqrt{(Q_{11}z)^2 - 1}}]\frac{1}{Q_{11}z + \sqrt{(Q_{11}z)^2 - 1}}$$

$$= \log(Q_{11}z + \sqrt{(Q_{11}z)^2 - 1}) - \log(Q_{11}) + o(\epsilon_m)$$

$$+ (\frac{245}{3}\epsilon_m Q_{11} + \frac{2}{3}\epsilon_8 + \epsilon_7) \cdot [1 + \frac{Q_{11}z}{\sqrt{(Q_{11}z)^2 - 1}}]\frac{1}{Q_{11}z + \sqrt{(Q_{11}z)^2 - 1}}$$

$$= \log(z + \sqrt{z^2 - Q_{11}^{-2}}) + o(\epsilon_m)$$

$$+ (\frac{245}{3}\epsilon_m Q_{11} + \frac{2}{3}\epsilon_8 + \epsilon_7) \cdot [1 + \frac{Q_{11}z}{\sqrt{(Q_{11}z)^2 - 1}}]\frac{1}{Q_{11}z + \sqrt{(Q_{11}z)^2 - 1}}$$

$$= (\frac{245}{3}C_0 E(C_0) + \frac{5}{3})\epsilon_m \cdot [1 + \frac{Q_{11}z}{\sqrt{(Q_{11}z)^2 - 1}}]\frac{1}{Q_{11}z + \sqrt{(Q_{11}z)^2 - 1}} + o(\epsilon_m)$$

$$+ \log(z + \sqrt{z^2 - Q_{11}^{-2}})$$

Hence, we get the error to be

$$\delta_2 = |\epsilon_1 \log(z + \sqrt{z^2 - Q_{11}^{-2}}) + \epsilon_2 + o(\epsilon_m)$$

$$+ (\frac{245}{3}C_0 E(C_0) + \frac{5}{3})\epsilon_m \cdot [1 + \frac{Q_{11}z}{\sqrt{(Q_{11}z)^2 - 1}}]\frac{1}{Q_{11}z + \sqrt{(Q_{11}z)^2 - 1}}|$$

$$\le (\frac{245}{3}C_0 E(C_0) + \frac{5}{3}) \cdot [1 + \frac{Q_{11}z}{\sqrt{(Q_{11}z)^2 - 1}}] + 1)\epsilon_m + (\log(C_0 + \sqrt{C_0^2 - 1})$$

$$- \log(Q_{11}) + o(\epsilon_m)$$

$$\le [\log(C_0 + \sqrt{C_0^2 - 1}) - \log(Q_{11}) + (\frac{245}{3}C_0 E(C_0) + \frac{8}{3}) \cdot (1 + \frac{45}{8\tanh(d)})]\epsilon_m$$

$$+ o(\epsilon_m)$$

$$\le [\log(2C_0) + (\frac{245}{3}C_0 E(C_0) + \frac{8}{3}) \cdot (1 + \frac{45}{8\tanh(d)})]\epsilon_m + o(\epsilon_m)$$

All in all, we can get the total error of computing distance

$$\delta = \delta_Q + \delta_2 = \log(Q_{11})\epsilon_m + \frac{1}{2}\epsilon_m + o(\epsilon_m) + \delta_2$$

Because $\log(Q_{11}) \le d$, then we get that if $Q_{11}z \ge C_0$,

$$\delta \le d\epsilon_m + (\frac{1825}{432} + \frac{245C_0}{3} + \log 70/3)\epsilon_m + o(C_0^{-2}) + o(\epsilon_m)$$

if $Q_{11}z \le C_0$,

$$\delta \le d\epsilon_m + [\frac{1}{2} + \log(2C_0) + (\frac{245}{3}C_0 E(C_0) + \frac{8}{3}) \cdot (1 + \frac{45}{8\tanh(d)})]\epsilon_m + o(\epsilon_m)$$

$\square$

Also, in the same way, we give the error for computing gradient:

**Theorem 7.** *The worst case error for computing gradient of distance in Lorentz model using float is*

$$
|\nabla_x d_l^{fl}(fl(x), fl(y)) - \nabla_x d_l(x, y)| = \frac{2 \cosh\left(d(x, O)\right) \cosh\left(d(y, O)\right)}{\sinh\left(d(x, y)\right) \tanh\left(d(x, y)\right)} \epsilon_m \nabla_x d
$$
$$
+ \frac{3}{4 \tanh^2\left(d(x, y)\right)} \epsilon_m \nabla_x d + \frac{1}{2} \epsilon_m \nabla_x d + o(\epsilon_m)
$$

*the error of computing gradient of distance in L-tiling model using float is*

$$
|\nabla_u d_{lt}^{fl} - \nabla_u d_{lt}|_1 = \begin{cases} [(B_1(C_0) + B_2(C_0) \exp(d)) |\nabla_x d|_1 + B_3(C_0) \exp(d)] \epsilon_m + o(\epsilon_m C_0^{-1}), \\ \qquad\qquad\qquad\qquad\qquad\qquad\qquad\qquad\qquad\qquad\qquad d \geq C_0 \\ [\frac{B_4(C_0)}{\tanh(d)} + (\frac{B_5(C_0)}{\tanh^2(d)} + \frac{B_6(C_0)}{\tanh(d)} + B_7(C_0)) |\nabla_x d|_1] \epsilon_m + o(\epsilon_m), \\ \qquad\qquad\qquad\qquad\qquad\qquad\qquad\qquad\qquad\qquad\qquad d \leq C_0 \end{cases}
$$

*where $d$ is the real distance between two points, $B_i(C_0)$ are constants only depends on $C_0$, $B_f$ is a fixed constant, $d^{fl}$ means that inside computation like multiplication are performed with machine error $\epsilon_m$.*

**Remark**: Similar to the worst case error of computing distance, the worst case error of computing gradient in Lorentz model using float is dominated by $d(x, O), d(y, O)$, this will also cause the "NaN" problem when two points are far away from the origin. In $L$-tiling model, the gradient error only depends on the gradient itself, i.e., also $\tanh$ term is bounded, which controls the error and solves the "NaN" problem.

*Proof.* We consider the Lorentz model at first, the gradient is

$$
\nabla_x d = \frac{My}{\sqrt{(x^T My)^2 - 1}},
$$

then we have the error to be

$$
|flc(\nabla_x d) - \nabla_x d|
$$
$$
= |(1 + \epsilon_1) \frac{(1 + \epsilon_2) My}{(1 + \epsilon_3)\sqrt{(1 + \epsilon_4)[(1 + \epsilon_5)(x^T My + \delta_z)^2 - 1]}} - \frac{My}{\sqrt{(x^T My)^2 - 1}}|
$$
$$
= \frac{\epsilon_m My}{2\sqrt{(x^T My)^2 - 1}} + \frac{\delta_z (x^T My) My}{((x^T My)^2 - 1)^{3/2}} + \frac{3\epsilon_m (x^T My)^2 My}{4((x^T My)^2 - 1)^{3/2}}
$$
$$
= \frac{1}{2} \epsilon_m \nabla_x d + \frac{3}{4 \tanh^2\left(d(x, y)\right)} \epsilon_m \nabla_x d + \frac{2 \cosh\left(d(x, O)\right) \cosh\left(d(y, O)\right)}{\sinh\left(d(x, y)\right) \tanh\left(d(x, y)\right)} \epsilon_m \nabla_x d + o(\epsilon_m)
$$

Here flc means calculating with float arithmetic. As for the gradient error in $L$-tiling model, note that the gradient is

$$
\nabla_u d_{lt}((U, u), (V, v)) = \frac{\nabla h(u)^T L^{-T} \hat{Q} L^{-1} h(v)}{\sqrt{z^2 - Q_{11}^{-2}}}
$$

where $\nabla h(u) = \left[ \frac{u}{\sqrt{1+||u||^2}}, \; I \right]$. First, consider

$$\text{flc}(\sqrt{(z+\delta_1)^2 - Q_{11}^{-2}})$$

$$=(1+\epsilon_1)\sqrt{(1+\epsilon_2)((1+\epsilon_3)(z+\frac{245}{3}\epsilon_m)^2 - (1+\epsilon_4)Q_{11}^{-2})}$$

$$=(1+\epsilon_1)(1+\epsilon_2/2)\sqrt{((1+\epsilon_3)(z+\frac{245}{3}\epsilon_m)^2 - (1+\epsilon_4)Q_{11}^{-2})} + o(\epsilon_m)$$

$$=(1+\frac{2}{3}\epsilon_5)\sqrt{(z+\frac{245}{3}\epsilon_m)^2 - Q_{11}^{-2} + \epsilon_6((z+\frac{245}{3}\epsilon_m)^2 + Q_{11}^{-2})} + o(\epsilon_m)$$

$$=(1+\frac{2}{3}\epsilon_5)[\sqrt{z^2 - Q_{11}^{-2} + \epsilon_6(z^2 + Q_{11}^{-2})} + \frac{245\epsilon_m z}{3\sqrt{z^2 - Q_{11}^{-2} + \epsilon_6(z^2 + Q_{11}^{-2})}}] + o(\epsilon_m)$$

$$=\sqrt{z^2 - Q_{11}^{-2} + \epsilon_6(z^2 + Q_{11}^{-2})} + \frac{245\epsilon_m z}{3\sqrt{z^2 - Q_{11}^{-2} + \epsilon_6(z^2 + Q_{11}^{-2})}}$$

$$+ \frac{2}{3}\epsilon_5\sqrt{z^2 - Q_{11}^{-2} + \epsilon_6(z^2 + Q_{11}^{-2})} + o(\epsilon_m)$$

$$=\sqrt{z^2 - Q_{11}^{-2} + \epsilon_6(z^2 + Q_{11}^{-2})} + \frac{245\epsilon_m z}{3\sqrt{z^2 - Q_{11}^{-2}}} + \frac{2}{3}\epsilon_5\sqrt{z^2 - Q_{11}^{-2}} + o(\epsilon_m)$$

$$=\sqrt{z^2 - Q_{11}^{-2}} + \frac{\epsilon_6(z^2 + Q_{11}^{-2})}{2\sqrt{z^2 - Q_{11}^{-2}}} + \frac{245\epsilon_m z}{3\sqrt{z^2 - Q_{11}^{-2}}} + \frac{2}{3}\epsilon_5\sqrt{z^2 - Q_{11}^{-2}} + o(\epsilon_m)$$

In the same way, we divide this into two cases, if $Q_{11}z \geq C_0$, then the error of this term is

$$\delta_4 = |\mathbf{fl}(\sqrt{(z+\delta_z)^2 - Q_{11}^{-2}}) - \sqrt{z^2 - Q_{11}^{-2}}|$$

$$=|\frac{\epsilon_6(z^2 + Q_{11}^{-2})}{2\sqrt{z^2 - Q_{11}^{-2}}} + \frac{245\epsilon_m z}{3\sqrt{z^2 - Q_{11}^{-2}}} + \frac{2}{3}\epsilon_5\sqrt{z^2 - Q_{11}^{-2}} + o(\epsilon_m)|$$

$$=|\frac{\epsilon_6 z}{2}(1 + \frac{3}{2(Q_{11}z)^2}) + \frac{245}{3}\epsilon_m(1 + \frac{1}{2(Q_{11}z)^2}) + \frac{2}{3}\epsilon_5\sqrt{z^2 - Q_{11}^{-2}}$$

$$+ O(\epsilon_m(Q_{11}z)^{-4}) + o(\epsilon_m)| \leq E_1\epsilon_m + O(\epsilon_m C_0^{-2}) + o(\epsilon_m)$$

Also, if $Q_{11}z \leq C_0$, then the error of this term is

$$\delta_4 = |\mathbf{fl}(\sqrt{(z+\delta_1)^2 - Q_{11}^{-2}}) - \sqrt{z^2 - Q_{11}^{-2}}|$$

$$=|\frac{\epsilon_6(z^2 + Q_{11}^{-2})}{2\sqrt{z^2 - Q_{11}^{-2}}} + \frac{245\epsilon_m z}{3\sqrt{z^2 - Q_{11}^{-2}}} + \frac{2}{3}\epsilon_5\sqrt{z^2 - Q_{11}^{-2}} + o(\epsilon_m)|$$

$$\leq \frac{E_2}{\tanh(d)}\epsilon_m + E_3\epsilon_m + o(\epsilon_m)$$

For the numerator term $z_p = \nabla h(u)^T L^{-T}\hat{Q}L^{-1}h(v)$, where $\nabla h(u) = \left[ \frac{u}{\sqrt{1+||u||^2}}, \; I \right]$, also because $|z_p|$ is bounded, then we can easily get that $\|\mathbf{fl}(z_{pi}) - z_{pi}\| \leq E_4\epsilon_m + o(\epsilon_m)$. Hence, we

have

$$\nabla_u d_{lt}((U, u), (V, v))_i = (1 + \epsilon_1) \frac{z_{pi} + E_4 \epsilon_m}{\sqrt{z^2 - Q_{11}^{-2}} + \delta_4}$$

$$= (1 + \epsilon_1)[\frac{z_{pi} + E_4 \epsilon_m}{\sqrt{z^2 - Q_{11}^{-2}}} - \frac{z_{pi} \delta_4}{z^2 - Q_{11}^{-2}}]$$

$$= \frac{z_{pi} + E_4 \epsilon_m}{\sqrt{z^2 - Q_{11}^{-2}}} + \frac{z_{pi} \delta_4}{z^2 - Q_{11}^{-2}} + \frac{z_{pi} \epsilon_1}{\sqrt{z^2 - Q_{11}^{-2}}}$$

Then, the error of gradient is

$$\delta_{gi} = \frac{E_4 \epsilon_m}{\sqrt{z^2 - Q_{11}^{-2}}} + \frac{z_{pi} \delta_4}{z^2 - Q_{11}^{-2}} + \frac{z_{pi} \epsilon_1}{\sqrt{z^2 - Q_{11}^{-2}}}$$

$$= \frac{E_4 \epsilon_m}{\sqrt{z^2 - Q_{11}^{-2}}} + \frac{\delta_4 \nabla_u d_{lti}}{\sqrt{z^2 - Q_{11}^{-2}}} + \epsilon_1 \nabla_u d_{lti}$$

So using this formula, we can get that if $Q_{11} z \geq C_0$, then the error of this term is

$$\delta_{gi} \leq E_4 Q_{11} \epsilon_m C_0^{-1} + [\delta_4 Q_{11} C_0^{-1} + \epsilon_m] \nabla_u d_{lti} + o(\epsilon_m C_0^{-1})$$

$$\leq E_4 Q_{11} \epsilon_m C_0^{-1} + [E_1 Q_{11} C_0^{-1} + 1] \epsilon_m \nabla_u d_{lti} + o(\epsilon_m C_0^{-1})$$

$$\leq E_4 \epsilon_m C_0^{-1} \exp(d) + [E_1 C_0^{-1} \exp(d) + 1] \epsilon_m \nabla_u d_{lti} + o(\epsilon_m C_0^{-1})$$

If $Q_{11} z \leq C_0$, then the error of this term is

$$\delta_{gi} \leq \frac{E_4}{\tanh(d)} \epsilon_m + [\frac{\delta_4}{\tanh(d)} + \epsilon_m] \nabla_u d_{lti}$$

$$\leq \frac{E_4}{\tanh(d)} \epsilon_m + [\frac{E_2}{\tanh^2(d)} + \frac{E_3}{\tanh(d)} + 1] \epsilon_m \nabla_u d_{lti} + o(\epsilon_m)$$

All in all, if $Q_{11} z \geq C_0$, then the error of gradient is

$$|\delta_g|_1 \leq 3 E_4 \epsilon_m C_0^{-1} \exp(d) + [E_1 C_0^{-1} \exp(d) + 1] \epsilon_m |\nabla_u d_{lt}|_1 + o(\epsilon_m C_0^{-1})$$

if $Q_{11} z \leq C_0$, then

$$|\delta_g|_1 \leq \frac{3 E_4}{\tanh(d)} \epsilon_m + [\frac{E_2}{\tanh^2(d)} + \frac{E_3}{\tanh(d)} + 1] \epsilon_m |\nabla_u d_{lt}|_1 + o(\epsilon_m)$$

$\square$

**Error for Computing in $H$-tiling Model**  Here we provide the error of computing in $H$-tiling model.

**Theorem 8.** *The error of computing distance in $H$-tiling model using float is*

$$|d_h^{fl}((j_1, \boldsymbol{k}_1, fl(\boldsymbol{z}_1)), (j_2, \boldsymbol{k}_2, fl(\boldsymbol{z}_2))) - d_h((j_1, \boldsymbol{k}_1, \boldsymbol{z}_1), (j_2, \boldsymbol{k}_2, \boldsymbol{z}_2))|$$

$$= [C_3(n)(j_2 - j_1) + C_4(n)d]\epsilon_m + (3e^{-d} + \frac{1}{e^d \sinh d})[C_1(n)2^{2j_2 - 2j_1}(1 + e^{d/2})^2 + C_2(n)]\epsilon_m$$

*where $d$ is the real distance between two points, $C_i(n)$ are constants only depends on $n$, $d^{fl}$ means that inside computation like multiplication are performed with machine error $\epsilon_m$.*

**Remark**: Similar to the worst case error of distance computation in $L$-tiling model, the worst case error in $H$-tiling model using float only depends on the distance itself, rather than how far points are from the origin, and hence controls the error and solves the "NaN" problem.

*Proof.* Firstly, note the distance is

$$d = (2s + j_1 - j_2)\log(2) + \log(2^{-2s - j_1 + j_2} + X + \sqrt{X^2 + 2^{1 - 2s - j_1 + j_2}X}),$$

where

$$X = \frac{\|I + 2^{-s}z_1 - 2^{j_2 - j_1 - s}z_2\|^2}{2z_{1n}z_{2n}}, \quad I = 2^{-s}(k_1 - 2^{j_2 - j_1}k_2)$$

Note that $\|I\| \leq 1, \|z_1\|, \|z_2\| \leq \sqrt{n + 3}$, then we can be bound $X$ in following way:

$$
\begin{aligned}
X &= \frac{\|I + 2^{-s}z_1 - 2^{j_2 - j_1 - s}z_2\|^2}{2z_{1n}z_{2n}} \\
&\leq \frac{\|I\|^2 + 2^{-2s}\|z_1\|^2 + 2^{2j_2 - 2j_1 - 2s}\|z_2\|^2}{2z_{1n}z_{2n}} \\
&\quad + \frac{2^{1 - 2s}\|z_1\|\|I\| + 2^{1 + 2j_2 - 2j_1 - 2s}\|z_2\|\|I\| + 2^{1 - 2s + j_2 - j_1}\|z_1\|\|z_2\|}{2z_{1n}z_{2n}} \\
&\leq \frac{1 + 2^{-2s}(n + 3) + 2^{2j_2 - 2j_1 - 2s}(n + 3)}{2z_{1n}z_{2n}} \\
&\quad + \frac{2^{1 - 2s}\sqrt{n + 3} + 2^{1 + 2j_2 - 2j_1 - 2s}\sqrt{n + 3} + 2^{1 - 2s + j_2 - j_1}(n + 3)}{2} \\
&\leq \frac{1}{2} + 2^{-2s - 1}(n + 3)(1 + 2^{2j_2 - 2j_1} + 2^{1 + j_2 - j_1}) + 2^{-2s}(1 + 2^{2j_2 - 2j_1})\sqrt{n + 3} \\
&\leq \frac{1}{2} + 2^{-2s + 1}((n + 3) + \sqrt{n + 3}) \leq 1 + 2^{-2s + 2}(n + 2)
\end{aligned}
$$

now consider the error for computing $I$

$$|\hat{I}_i - I_i| = |(1 + \epsilon_1)2^{-s}(k_{1i} - 2^{j_2 - j_1}k_{2i}) - I_i| \leq \epsilon_m|I_i| = \delta_{i,1}$$

here denote $X_u = I + 2^{-s}z_1 - 2^{j_2 - j_1 - s}z_2$, then

$$
\begin{aligned}
|\hat{X_{ui}} - X_{ui}| &= |[(I_i + \delta_{i,1} + (1 + \epsilon_1)2^{-s}z_{1i})(1 + \epsilon) - (1 + \epsilon_2)2^{j_2 - j_1 - s}z_{2i}](1 + \epsilon) - X_{ui}| \\
&= |[I_i + 2^{-s}z_{1i} + \delta_{i,1} + \epsilon_1 2^{-s}z_{1i} + \epsilon(I_i + 2^{-s}z_{1i}) - (1 + \epsilon_2)2^{j_2 - j_1 - s}z_{2i}](1 + \epsilon) - X_{ui}| \\
&\leq |[I_i + 2^{-s}z_{1i} - 2^{j_2 - j_1 - s}z_{2i} + \delta_{i,1} + \epsilon_m(I_i + 2^{1 - s}z_{1i} + 2^{j_2 - j_1 - s}z_{2i})](1 + \epsilon) - X_{ui}| \\
&\leq |\epsilon_m X_{ui} + \delta_{i,1} + \epsilon_m(I_i + 2^{1 - s}z_{1i} + 2^{j_2 - j_1 - s}z_{2i})| \\
&\leq \epsilon_m(|X_{ui}| + 2|I_i| + 2^{1 - s}|z_{1i}| + 2^{j_2 - j_1 - s}|z_{2i}|) = \delta_{i,2}
\end{aligned}
$$

next, make use of float arithmetic to get the norm of $X_u$, the error becomes

$$|\|\hat{X}_u\|^2 - \|X_u\|^2|$$

$$\leq |(1 + n\epsilon_m)\|X_u + \delta_2\|^2 - \|X_u\|^2| = |n\epsilon_m\|X_u\|^2 + \sum_{i=1}^{n}(\delta_{i,2}^2 + 2X_{ui}\delta_{i,2})|$$

$$\leq |n\epsilon_m\|X_u\|^2 + \sum_{i=1}^{n} 2X_{ui}\delta_{i,2} + o(\epsilon_m)|$$

$$= \sum_{i=1}^{n}(4|I_i| + 2^{2-s}|z_{1i}| + 2^{1+j_2-j_1-s}|z_{2i}|)\epsilon_m|X_{ui}| + (n+2)\epsilon_m\|X_u\|^2 + o(\epsilon_m)$$

$$\leq (4\|z_1\| + 2^{1+j_2-j_1}\|z_2\|)\epsilon_m 2^{-s}\|X_u\| + 4\epsilon_m\|I\|\|X_u\| + (n+2)\epsilon_m\|X_u\|^2 + o(\epsilon_m) = \delta_3$$

Note that $X = \frac{X_u^2}{2z_{1n}z_{2n}}$, then we bound the error of computing $X$ here:

$$|\hat{X} - X| = |\frac{\|\hat{X}_u\|^2}{2z_{1n}z_{2n}(1+2\epsilon)} - X| = |\frac{\|X_u\|^2 + \delta_3}{2z_{1n}z_{2n}}(1-2\epsilon) - X| \leq |\frac{\delta_3}{2z_{1n}z_{2n}} - 2\epsilon X|$$

$$\leq 2|\epsilon_m X| + |\frac{\delta_3}{2z_{1n}z_{2n}}| \leq 2|\epsilon_m X| + |\frac{\delta_3}{2}|$$

$$\leq 2|\epsilon_m X| + (2\|z_1\| + 2^{j_2-j_1}\|z_2\|)\epsilon_m 2^{-s}\|X_u\| + 2\epsilon_m\|I\|\|X_u\|$$

$$+ (n/2 + 1)\epsilon_m\|X_u\|^2 + o(\epsilon_m)$$

$$\leq 2\epsilon_m X + (2 + 2^{j_2-j_1})\sqrt{n+3}\epsilon_m 2^{1-s}\sqrt{2X} + 4\epsilon_m\sqrt{2X} + 4(n+2)\epsilon_m X + o(\epsilon_m)$$

$$\leq 2\epsilon_m X + 2^{2+j_2-j_1-s}\epsilon_m\sqrt{2(n+3)X} + 4\epsilon_m\sqrt{2X} + 4(n+2)\epsilon_m X + o(\epsilon_m)$$

$$\leq 2\epsilon_m(1 + 2^{-2s+2}(n+2)) + 2^{3+j_2-j_1-s}\epsilon_m\sqrt{(n+3)(1+2^{-2s+2}(n+2))}$$

$$+ 6\epsilon_m\sqrt{1 + 2^{-2s+2}(n+2)} + 4(n+2)(1 + 2^{-2s+2}(n+2))\epsilon_m = \delta_4$$

After having these errors, consider the computation error for the second $\log$ in distance:

$$\log(2^{-2s-j_1+j_2} + (1+\epsilon)(\hat{X} + \sqrt{1+\epsilon}\sqrt{(1+\epsilon)\hat{X}^2 + 2^{1-2s-j_1+j_2}\hat{X}})) + \log(1+\epsilon)$$

$$= \log(2^{-2s-j_1+j_2} + (1+\epsilon)(\hat{X} + \sqrt{X^2 + 2^{1-2s-j_1+j_2}X} + \frac{1}{2}\epsilon\sqrt{X^2 + 2^{1-2s-j_1+j_2}X}$$

$$+ \frac{\delta_4(2^{-1-2s-j_1+j_2} + X/2)}{\sqrt{X^2 + 2^{1-2s-j_1+j_2}X}})) + \epsilon$$

$$= \log(2^{-2s-j_1+j_2} + X + \delta_4 + \sqrt{X^2 + 2^{1-2s-j_1+j_2}X} + \frac{1}{2}\epsilon\sqrt{X^2 + 2^{1-2s-j_1+j_2}X}$$

$$+ \frac{\delta_4(2^{-1-2s-j_1+j_2} + X/2)}{\sqrt{X^2 + 2^{1-2s-j_1+j_2}X}})) + \epsilon(X + \sqrt{X^2 + 2^{1-2s-j_1+j_2}X})) + \epsilon$$

$$= \log(2^{-2s-j_1+j_2} + X + \sqrt{X^2 + 2^{1-2s-j_1+j_2}X}) + \epsilon$$

$$+ \frac{\delta_4 + \frac{1}{2}\epsilon\sqrt{X^2 + 2^{1-2s-j_1+j_2}X} + \frac{\delta_4(2^{-1-2s-j_1+j_2}+X/2)}{\sqrt{X^2+2^{1-2s-j_1+j_2}X}})) + \epsilon(X + \sqrt{X^2 + 2^{1-2s-j_1+j_2}X})}{2^{-2s-j_1+j_2} + X + \sqrt{X^2 + 2^{1-2s-j_1+j_2}X}}$$

All in all, we combine all errors to derive the error for distance computation:

$$
\begin{aligned}
\delta_0 =& (2s+j_1-j_2)\log(2)\epsilon + \log(1+\epsilon)\\
&+\frac{\delta_5 + \frac{1}{2}\epsilon\sqrt{X^2+2^{1-2s-j_1+j_2}X}+\frac{\delta_5(2^{-1-2s-j_1+j_2}+X/2)}{\sqrt{X^2+2^{1-2s-j_1+j_2}X}}))+\epsilon(X+\sqrt{X^2+2^{1-2s-j_1+j_2}X})}{2^{-2s-j_1+j_2}+X+\sqrt{X^2+2^{1-2s-j_1+j_2}X}}\\
=&(2s+j_1-j_2)\log(2)\epsilon+\epsilon\\
&+\frac{\delta_5+\frac{1}{2}\epsilon\sqrt{X^2+2^{1-2s-j_1+j_2}X}+\frac{\delta_5(2^{-1-2s-j_1+j_2}+X/2)}{\sqrt{X^2+2^{1-2s-j_1+j_2}X}}))+\epsilon(X+\sqrt{X^2+2^{1-2s-j_1+j_2}X})}{2^{-2s-j_1+j_2}+X+\sqrt{X^2+2^{1-2s-j_1+j_2}X}}\\
=&(2s+j_1-j_2)\log(2)\epsilon+\epsilon+\frac{\delta_5(\frac{3}{2}+\frac{2^{-1-2s-j_1+j_2}}{\sqrt{X^2+2^{1-2s-j_1+j_2}X}})+\frac{3}{2}\epsilon\sqrt{X^2+2^{1-2s-j_1+j_2}X}+\epsilon X}{2^{-2s-j_1+j_2}+X+\sqrt{X^2+2^{1-2s-j_1+j_2}X}}\\
=&(2s+j_1-j_2)\log(2)\epsilon+\epsilon+\frac{3}{2}\epsilon+2^{2s+j_1-j_2}\frac{(\frac{3}{2}+\frac{1}{2\sinh d})}{\cosh d+\sinh d}\delta_5\\
=&(2s+j_1-j_2)\log(2)\epsilon+\frac{5}{2}\epsilon+2^{2s+j_1-j_2-1}\frac{(3+\frac{1}{\sinh d})}{\cosh d+\sinh d}[2(1+2^{-2s+2}(n+2))\\
&+2^{3+j_2-j_1-s}\sqrt{(n+3)(1+2^{-2s+2}(n+2))}+6\sqrt{1+2^{-2s+2}(n+2)}\\
&+4(n+2)(1+2^{-2s+2}(n+2))]\epsilon_m
\end{aligned}
$$

We simplify the constants with $C_i(n)$, also note that $k_1 - 2^{j_2-j_1}k_2$ is an integer vector, then $s = \lceil\log_2(\|k_1 - 2^{j_2-j_1}k_2\|^2)/2\rceil \geq 0$, also we have $j_2 \geq j_1$, then the error for distance computation is

$$
\begin{aligned}
\delta_0 \leq& (3+2s+j_1-j_2)\epsilon+2^{2s+j_1-j_2-1}\frac{(3+\frac{1}{\sinh d})}{\cosh d+\sinh d}[C_1(n)+2^{-2s}C_2(n)+2^{j_2-j_1-2s}C_3(n)]\epsilon_m\\
\leq& (3+2s+j_1-j_2)\epsilon+\frac{(3+\frac{1}{\sinh d})}{\cosh d+\sinh d}[2^{2s+j_1-j_2}C_1(n)+2^{j_1-j_2}C_2(n)+C_3(n)]\epsilon_m\\
\leq& (3+2s)\epsilon+\frac{(3+\frac{1}{\sinh d})}{\cosh d+\sinh d}[2^{2s}C_1(n)+C_2(n)]\epsilon_m\\
=& \lceil\log_2(\|k_1-2^{j_2-j_1}k_2\|^2)\rceil\epsilon_m+\frac{(3+\frac{1}{\sinh d})}{\cosh d+\sinh d}[C_1(n)\|k_1-2^{j_2-j_1}k_2\|^2+C_2(n)]\epsilon_m
\end{aligned}
$$

To bound the norm in the distance error, consider

$$
\begin{aligned}
\|2^{j_1}k_1-2^{j_2}k_2\| =& \|2^{j_1}z_1-2^{j_2}z_2+2^{j_1}k_1-2^{j_2}k_2+2^{j_2}z_2-2^{j_1}z_1\|\\
\leq& \|2^{j_1}z_1-2^{j_2}z_2+2^{j_1}k_1-2^{j_2}k_2\|+\|2^{j_2}z_2-2^{j_1}z_1\|\\
\leq& 2^{\frac{j_1+j_2+1}{2}}\sqrt{(\cosh d-1)z_{1n}z_{2n}}+2^{j_2}\|z_2\|+2^{j_1}\|z_1\|\\
\leq& 2^{\frac{j_1+j_2}{2}+2}\sinh(d/2)+(2^{j_2}+2^{j_1})\sqrt{n+3}
\end{aligned}
$$

then scale it by $2^{-j_1}$, we have

$$
\|k_1-2^{j_2-j_1}k_2\| \leq 2^{\frac{j_2-j_1}{2}+2}\sinh(d/2)+(2^{j_2-j_1}+1)\sqrt{n+3}
$$

Therefore, we bound the distance computation error as follows:

$$
\delta_0 \leq [C_3(n)(j_2-j_1)+C_4(n)d]\epsilon_m+(3e^{-d}+\frac{1}{e^d\sinh d})[C_1(n)2^{2j_2-2j_1}(1+e^{d/2})^2+C_2(n)]\epsilon_m
$$

$\square$