[Reviews · NeurIPS 2019]

Reviewer 1



This paper provides a novel and interesting solution to the problem of numerical instability in ML based methods on hyperbolic manifolds using a group based tiling approach. The work is well supported by experiments and the mathematical statements are backed up by extensive proofs in the supplementary material. The work is generally very clear considering the complexity of the underlying approach. In places, however, the writing seems rushed and would benefit from the authors reviewing the language. I have highlighted several typos in the improvements, but there are likely more. There are also some inconsistencies in the bibliography. The major limitation is that it does not apply to dimensions greater than 6, which are required by most existing work in this field. The authors mitigate this problem in Section 6, so that it is my belief that the work is still significant, though less so than were the group based tilings to be readily applicable to many existing models.

Reviewer 2



The paper touches an issue that is very important and most likely the reason why hyperbolic embeddings have not been adopted widely. From my experience, hyperbolic embeddings sometimes have catastrophic results compared with competing methods. This is because of numerical instabilities. The paper is very well written with a lot of theoretical and empirical results. The solutions the authors provide is theoretically proven and very well documented. The experiments are also sufficient and realistic and they prove the point. The only problem of this paper is that it has too much information that most of the NeurIPS audience would not be able to follow, as it is not familiar with a lot of interesting mathematical terms. I spent significant time to go through the references. Given the complexity of the solution, the authors have done a good job explaining some mathematical concepts, but they definitely needed more space. The supplementary material is 16 pages! line 45: A schematic would explain the idea better reference [4] : “In In Flavors of geometry “ repeated “In” line 71:” Since for hierarchical data such as a tree with branching factor b, the number of leaf nodes increases exponentially as the number of levels increases”. I have seen this explanation in several papers but I think it is not correct. Also, this argument is fundamentaly wrong as it compares an infinite space with a finite space. The number of the leafs of a hierarchical space grows exponentially but remains finite. The area of a disk contains infinite points. Technically speaking the unit circle can fit the universe. The reason why euclidean space is different and it is described very well on this paper: As described in this paper https://papers.nips.cc/paper/5971-space-time-local-embeddings.pdf “...The maximum number of points which can share a common nearest neighbor is limited 2 for 1-dimensional spaces, 5 for 2-dimensional spaces and so on.While such centralized structures do exist in real data d-dimensional spaces can at most embed (d + 1) points with uniform pair-wise similarities.” Also the triangular inequality imposes more constraints. See the references: K. Zeger and A. Gersho. How many points in Euclidean space can have a common nearest neighbor? In International Symposium on Information Theory, page 109, 1994. L. van der Maaten and G. E. Hinton. Visualizing non-metric similarities in multiple maps. Machine Learning, 87(1):33–55, 2012. I suggest reading this blog https://networkscience.wordpress.com/2011/08/09/dating-sites-and-the-split-complex-numbers/ and also this paper: https://dl.acm.org/citation.cfm?id=2365942 line 86: “This suggests that the numerical model used for learning an embedding can have significant impact on its performance.” To me that doesn’t come as a logical consequence. The reference talks about the importance of curvature, flat, negative, positive. At least that is the main point. The title doesn’t address the numerical issues. line 91:” However, those models suffer from the precision and accuracy problem” We need a little bit of clarification here. Where the models compared with other embeddings and didn’t prove to be competitive in performance? Do the authors believe that the reason was numerical instability? If yes it would be nice to repeat these experiments with their solution and prove the point. line 131: The NeurIPS audience is not very familiar with Fuchsian groups. I had to do a lot of reading to understand how the tiling is done. That paragraph is too condensed line 146:”plane as a pair consists of “ change to “plane as a pair that consists of” line 162: “Importantly, any element of G can be represented in the form “ Is this a fact for all Fuchsian matrices? line 248:”Construct tilings from a set of isometries that is not a group.” This is an interesting idea. Can we have a reference for that? An example of isometries that are not groups would be useful here.

Reviewer 3



Originality: the method is very novel and creative, some of the best ML papers I recently read. Previous work is mostly well cited, with some exceptions of recent hyperbolic embeddings papers from recent NeurIPS/ ICLR / ICML conferences. Quality: As said above, the proposed method is usually technically sound and complete, e.g. section 4 and attached appendices, but has a few weaknesses or incomplete discussions in sections 5 and 6 that I will briefly touch below. The authors are typically honest about their work, with the small exception of Algorithm 2 (RSGD) which works only for restricted objectives that depend only on the hyperbolic distance function. My current grade is conditioned on clarifying this issue in the main text. Clarity: paper is clearly written in general. However, sections 5 and 6 and their appendices should be explained more extensively (I know the page limit is a problem, but at least the appendices should be clear). The experimental section is too small for an easy reproducibility of this method, so I am asking the authors to both publish the full code to reproduce all the reported results and to do a better job at explaining the training details (and hyperparameters). Significance: as said above, this is an important contribution to improve hyperbolic embeddings and reduce their inherent numerical errors. It would be hard for someone else to redo this important piece of research and its valuable theory. However, I believe there are still a few points that could be improved both in the write-up and method that would make the paper stronger.

[Author Response · NeurIPS 2019]

We thank the reviewers for useful and detailed feedback, which has helped us to improve our paper. We will release our code soon. First, we will respond to questions common among the reviewers, and then address individual concerns.

All reviewers asked about the higher-dimensional case. To clarify, any tiling with all-congruent tiles (even in higher dimensions) can be represented as a subset of the set of isometries. This subset being a group, as for L-tilings is desirable because the symmetries involved simplify the analysis and representation of the tiles. For H-tilings, the subset is not a group, and this can't be avoided in higher dimensions because of Coxeter's result [6]. Nevertheless H-tilings can still be used for learning, as we show in Section 6. Despite their inapplicability to high-dimensional spaces, L-tilings are still desirable in cases where we do want to learn/embed over 2D space or the Cartisian product of 2D spaces, as was done in Gu et al [14]. We will update our manuscript to clarify this point.

R2 and R3 correctly pointed out that currently our methods apply to loss functions that depend on distances. However, most embedding tasks such as word embedding and link prediction in networks, do depend on distances and can be computed efficiently with our methods. Extension to other loss functions will be future work.

Q1 from R1: Regarding line 114, "its curvature still . . ."? ——A1: We meant to say that the Riemannian metric of the model becomes large and poorly conditioned, and not the manifold curvature. We have fixed this.

Q2 from R2: In line 86, "This suggests that . . ." doesn't come as a logical consequence. ——A2: We misplaced this sentence; it was intended to refer to the comparison between Poincare and Lorenz models (two different numerical models) in [19], not to Gu et al [14].

Q3 from R2: Does the numerical instability affect the performance of models for different tasks mentioned in line 91? ——A3: We believe numerical stability is an issue for different tasks based on communications with authors of some cited papers. We plan to cut this claim and leave that evaluation to future work.

Q4 from R2: In line 162, "Importantly, any element ...", is this a fact for all Fuchsian matrices? ——A4: No, it's not generally true for Fuchsian groups, we choose the generators (integer matrix) in Definition 1 to make it happen.

Q5 from R3: Clarify the meaning of SGD and RSGD? ——A5: In this paper, SGD uses the Euclidean gradient within the model, while RSGD transforms the Euclidean gradient to a Riemannian gradient, and uses the exponential map on the manifold to update parameters. All models including baselines are trained with RSGD except that we specifically train a L-tiling model with SGD for comparison as mentioned in Line 282.

Q6 from R3: Explain algorithms 1,2 and the minimization of $W$ in algorithm 2, how computationally expensive it is (explain theoretically and empirically)? What are the training time? ——A6: As Theorem 3 and 6 states, algorithm 1 maps a point in the Lorentz model to a point $(U, u)$ in the L-tiling model, where $u$ is unique in $F$, so algorithm 1 outputs the solution $U$ of the minimization problem in algorithm 2. In the proof of algorithm 2, $W = U^{-1}V$ is a middle variable for convenience and different from the $W$ in the minimization, we will change the symbols and rewrite this part. The computational complexity of algorithm 1 is linear in the distance of the point from the origin as shown in Theorem 3. Empirically, for an existing embedding of Gr-QC dataset (4158 nodes), in which the largest and average absolute value are 2.05e+10 and 1.48e+07, it will take 0.92 seconds to solve all minimization problems. But for training, points are initialized near the Origin, the minimization problem is solved once the point is out of $F$, so typically it will finished within 3 steps. As for the training time, take the learning of 2D embeddings for Wordnet Verbs for example, same as released baselines' code, which trains the embedding on CPU, we trained 5 models for 1000 epochs, here is the time: L-tiling-SGD: 27079s, L-tiling-RSGD: 18028s, Lorentz: 7867s, Poincare: 20422s, H-tiling; 18388s.

Q7 from R3: How to translate from a given matrix U to the VBW encoding? ——A7: If $U$ is given, consider the point $(U, O)$ in the L-tiling model, which is $x = LUL^{-1}O$ in the Lorentz model, then we can map $x$ to $(U', u')$ with algorithm 1, where we choose a generator at each step, then we can store a generator order string from algorithm 1, with which we can reconstruct $U'$. Since each point in the Lorentz model will be mapped to a unique point in $F$, also $x$ can be mapped to $(U, O)$ and $(U', u')$, so $u' = O$. The question is whether $U = U'$, consider $LUL^{-1}O = LU'L^{-1}O$, which leads to $(LU'^{-1}UL^{-1} - I)O = 0$, as Line 37-39 in appendix shows, we prove that $LU'^{-1}UL^{-1} = I$, then $U = U'$. Hence, given $U$, we can get its generator order string, then we can get the VBW encoding accordingly.

Q8 from R3: Can you prove the statement in line 252: each square is isometric to every other square? ——A8: We've called them squares even though in the hyperbolic metric they bear no resemblance to squares. See page 95-98 of Cannon et al [4] for an introduction of these isometries and "squares" with a nice graph; We will more clearly reference this in our updated manuscript.

We will improve write-up and methods in the following way: add mathematical concepts like Fuchsian group into appendix, add a section about learning in appendix to explain more extensively of section 5 and 6. For experiments, add product of baseline models for dimensions 4 and 6, add confidence intervals to the results, add a training detail section in appendix. Also, fix some typos, statements and inconsistencies in the bibliography.

[Meta-Review · NeurIPS 2019]

There is broad agreement on the elegance and creativity of the proposed tiling-based methods. It is clear that this can address (practical) numerical problems, but also provide inspirations for understanding and using hyperbolic embeddings. While hyperbolic embeddings are still a bit of a niche topic, I feel that the paper contributes nicely to the non-trivial mathematics in this area and I am supportive to publish the paper at NeurIPS.